# Inhalable biohybrid microrobots: a non-invasive approach for lung treatment

Zhengxing Li [1,2,5], Zhongyuan Guo[1,5], Fangyu Zhang [1,5], Lei Sun[1,5], Hao Luan[1,2], Zheng Fang [1], Jeramy L. Dedrick[3], Yichen Zhang[2], Christine Tang[1], Audrey Zhu[1], Yiyan Yu [1], Shichao Ding [1], Dan Wang[1], An-Yi Chang [1], Lu Yin [1], Lynn M. Russell [3], Weiwei Gao [1], Ronnie H. Fang [1,4], Liangfang Zhang [1,2] ✉ & Joseph Wang [1,2] ✉

Amidst the rising prevalence of respiratory diseases, the importance of effective lung treatment modalities is more critical than ever. However, current drug delivery systems face significant limitations that impede their efficacy and therapeutic outcome. Biohybrid microrobots have shown considerable promise for active in vivo drug delivery, especially for pulmonary applications via intratracheal routes. However, the invasive nature of intratracheal administration poses barriers to its clinical translation. Herein, we report on an efficient non-invasive inhalation-based method of delivering microrobots to the lungs. A nebulizer is employed to encapsulate picoeukaryote algae microrobots within small aerosol particles, enabling them to reach the lower respiratory tract. Post nebulization, the microrobots retain their motility (~55 μm s⁻¹) to help achieve a homogeneous lung distribution and long-term retention exceeding five days in the lungs. Therapeutic efficacy is demonstrated in a mouse model of acute methicillin-resistant *Staphylococcus aureus* pneumonia using this pulmonary inhalation approach to deliver microrobots functionalized with platelet membrane-coated polymeric nanoparticles loaded with vancomycin. These promising findings underscore the benefits of inhalable biohybrid microrobots in a setting that does not require anesthesia, highlighting the substantial translational potential of this delivery system for routine clinical applications.

Microrobots have recently demonstrated considerable promise for enhancing therapeutic and diagnostic biomedical applications[1–3]. Depending on the in vivo organ or disease site being targeted, the administration methods of microrobots are tailored to optimize their specific functionalities and enhance therapeutic outcomes. For example, metallic microrobots have been administered orally as drug carriers to the gastrointestinal system[4,5], polymeric micropropellers have been delivered intravitreally for ocular drug delivery[6], lipid-shelled microbubbles have been introduced intravenously for therapeutic applications within the brain vasculature[7], and urease-powered mesoporous silica-based nanobots have been used for treating bladder cancer[8]. Considerable attention has recently been given to lung drug delivery due to the increasing prevalence of respiratory diseases[9,10]. Along these lines, biohybrid microrobots were delivered via

---

[1]Aiiso Yufeng Li Family Department of Chemical and Nano Engineering, University of California San Diego, La Jolla, CA, USA. [2]Program in Materials Science and Engineering, University of California San Diego, La Jolla, CA, USA. [3]Scripps Institution of Oceanography, University of California San Diego, La Jolla, CA, USA. [4]Division of Host-Microbe Systems and Therapeutics, Department of Pediatrics, University of California San Diego, La Jolla, CA, USA. [5]These authors contributed equally: Zhengxing Li, Zhongyuan Guo, Fangyu Zhang, Lei Sun. ✉e-mail: zhang@ucsd.edu; josephwang@ucsd.edu

intratracheal administration to treat acute pneumonia[11] and lung metastasis[12]. However, despite the benefits of direct delivery to the lungs, the intratracheal route is often hindered by its invasive nature and associated patient discomfort, which limits its broader clinical applicability[13,14].

Here, we report on a nebulizer system designed for the efficient pulmonary delivery of therapeutic microrobots. Inhalation-based drug delivery represents a highly attractive approach for treating lung diseases[15,16]. Non-invasive administration provides substantial benefits, including targeted delivery of therapeutic agents to the lungs along with reduced side effects, as well as improved patient comfort and safety, towards enhanced therapeutic outcomes and widespread clinical use[15,16]. Drug carriers can significantly enhance therapeutic efficacy by stabilizing active agents, increasing bioavailability, and facilitating targeted and controlled release directly to the affected pulmonary tissues[17,18]. Various micro-/nanoscale platforms, such as polymeric nanoparticles, liposomes, lipid nanoparticles, or micelles, have been used as carriers for inhalation-based treatment of lung diseases[15,19–23]. However, such static drug carriers commonly encounter various limitations post inhalation, including rapid clearance and elimination by alveolar macrophages, which compromise their therapeutic efficacy and increase the risk of systemic side effects[24,25]. The inherent active motility of microrobot drug carriers offers considerable promise for addressing these limitations. Furthermore, biohybrid microrobots, relying on biological micro-engines, offer significant advantages over their synthetic counterparts, which can be encumbered by limited fuel availability, restricted access to specific organs and tissues, and potential toxicity[26–31]. Combining the advantages of both inhalation-based delivery and biohybrid microrobots is thus expected to result in an effective platform for treating lung diseases.

In this study, the inhalable microrobot-based lung delivery system relies on the green algae *Micromonas pusilla*[32] as its actuator (denoted as 'algae robot') (Fig. 1a and Supplementary Fig. 1). Aerosol particle size is a critical factor, as it determines the ability of the nebulized droplets to reach the alveolar region effectively[33,34]. The small size of the algae robots (approximately 1–1.5 μm in diameter), enables them to be effectively encapsulated within aerosol particles, 98.6% of which are smaller than 10 μm, facilitating optimal inhalation into the lungs. This leads to a stable distribution and prolonged retention exceeding five days in the mouse lungs post nebulization, reflecting the ability of the algae robots to inhibit macrophage phagocytosis via their self-propulsion capabilities. Furthermore, vancomycin, as a model drug, is encapsulated into the platelet membrane-coated poly(lactic-*co*-glycolic acid) (PLGA) nanoparticles (PNPs)[35], which are subsequently conjugated onto the algae robot surface via click chemistry[36,37]. The resulting formulation (denoted as 'algae-PNP(Vanc)-robot') is thoroughly evaluated and demonstrates enhanced therapeutic efficacy when delivered by the nebulization approach to treat a murine model of methicillin-resistant *Staphylococcus aureus* (MRSA) infection.

## Results

### Characterization of the algae robot-containing aerosol particles

*M. pusilla* were cultivated in L1-Si medium and maintained in a 22 °C thermostatic incubator (Supplementary Fig. 2). Scanning electron microscope (SEM) imaging illustrated that the algae robot comprised an ~1.2 μm ellipsoid body and a single flagellum that provides motility (Fig. 1b and Supplementary Fig. 3a). In addition, the algae robot exhibited a strong autofluorescence signal in the Cy5 channel, which is attributed to the presence of chlorophylls associated with the photosynthetic apparatus within the chloroplasts[38] (Supplementary Fig. 3b, c). The algae robots retained their inherent motility in various biological media, such as 1 × PBS buffer and simulated lung fluid (SLF, Supplementary Table 1), highlighting their suitability for in vivo biomedical applications (Supplementary Fig. 4 and Supplementary Movie 1). Subsequently, the algae robots were transferred into PBS and

aerosolized using a jet nebulizer system that generated a stable aerosol flow (Supplementary Fig. 5 and Supplementary Movie 2). To facilitate visualization, the algae robot-containing aerosol particles were suspended in an oil phase, and it was observed that they maintained their motility within the droplets, indicating successful aerosol encapsulation (Fig. 1c, Supplementary Fig. 6 and Supplementary Movie 3). An aerodynamic size distribution analysis demonstrated that approximately 98.6% of the algae robot-containing aerosol particles were smaller than 10 μm, indicating their potential for inhalable therapeutic applications in the lungs[39] (Fig. 1d). The algae robots also retained their structural integrity (Supplementary Fig. 7a, b) and autofluorescence properties (Supplementary Fig. 7c, d) after encapsulation into the aerosol particles. We next sought to understand the impact of critical factors for nebulization such as algae robot loading concentration and the system air flow rate (Fig. 1e). When the system air flow rate was maintained at 6 L min⁻¹ and the algae robot loading was set at $1 \times 10^7$ mL⁻¹ or $1 \times 10^8$ mL⁻¹, it was observed that 73.89 ± 3.18 % and 73.77 ± 2.94 % of the algae robots retained their motility ($p = 0.9556$) post nebulization, respectively. Increasing the algae robot loading to $1 \times 10^9$ mL⁻¹ or $1 \times 10^{10}$ mL⁻¹ resulted in a decreased motile ratio of 56.44 ± 5.48 % or 39.85 ± 5.07 %, respectively. Using a fixed algae robot loading of $1 \times 10^7$ mL⁻¹ while increasing the system air flow rate from 4 L min⁻¹ to 10 L min⁻¹ also led to decreased motility from 86.13 % to 57.50 %. Overall, an algae robot loading of $1 \times 10^8$ mL⁻¹, combined with a system air flow rate of 4 L min⁻¹, was determined to be the optimal configuration and was used for further studies.

### Characterization of algae robot motility post nebulization

We further elucidated the influence of algae robot loading and system air flow rate on the post-nebulization motility of the algae robot in SLF. First, the aerodynamic diameter of aerosol particles at various algae robot loadings using a system air flow rate of 4 L min⁻¹ was measured. We observed that, as the algae robot loading increased, the size distribution of the aerosol particles became more dispersed (Fig. 2a). The increased loading also resulted in higher unencapsulated algae robot ratios (Supplementary Fig. 8), as well as a concomitant decrease in average speed from 58.46 ± 6.42 μm s⁻¹ to 45.46 ± 9.87 μm s⁻¹ (Fig. 2b and Supplementary Movie 4). This modest reduction in speed did not compromise the active functionality of biohybrid microrobots as compared to their static counterparts. Likewise, algae motility and viability were also negatively impacted, and it was determined that an algae robot loading of $1 \times 10^8$ mL⁻¹ provided the best balance between algae performance and loading (Fig. 2c and Supplementary Fig. 9). At this algae robot loading of $1 \times 10^8$ mL⁻¹, we proceeded to evaluate the impact of modulating the system air flow rate from 4 to 10 L min⁻¹ (Supplementary Fig. 10). As the flow rate increased, the size distribution of aerosol particles exhibited a pronounced left shift from ~5.2 μm to ~3.4 μm, indicating an increased likelihood of damage to the algae robot flagella (Fig. 2d). This was reflected in reductions of the average speed (from 58.07 ± 6.17 μm s⁻¹ to 47.96 ± 10.23 μm s⁻¹), motile ratio (from 84.84 % to 53.70 %), and viability (from 96.58 % to 68.54 %) of the algae post nebulization with increasing the system air flow rate (Fig. 2e, f, Supplementary Fig. 11 and Supplementary Movie 5). We next evaluated the impact of operation time on the performance of the nebulizer system. During the nebulization process, we collected samples at 5, 15, 30, 45, and 60 min for analysis. There was little change in the aerodynamic diameter of the aerosol particles collected at different time points (Fig. 2g). Moreover, the algae robot speed, motile ratio, and viability data all remained consistent for the duration of the study, validating the ability of the algae robot nebulizer system to be used for long durations (Fig. 2h, i and Supplementary Fig. 12). Representative motion trajectories over 1 s also indicated that the algae robots traveled comparable distances with similar trajectories after nebulization (Fig. 2j).

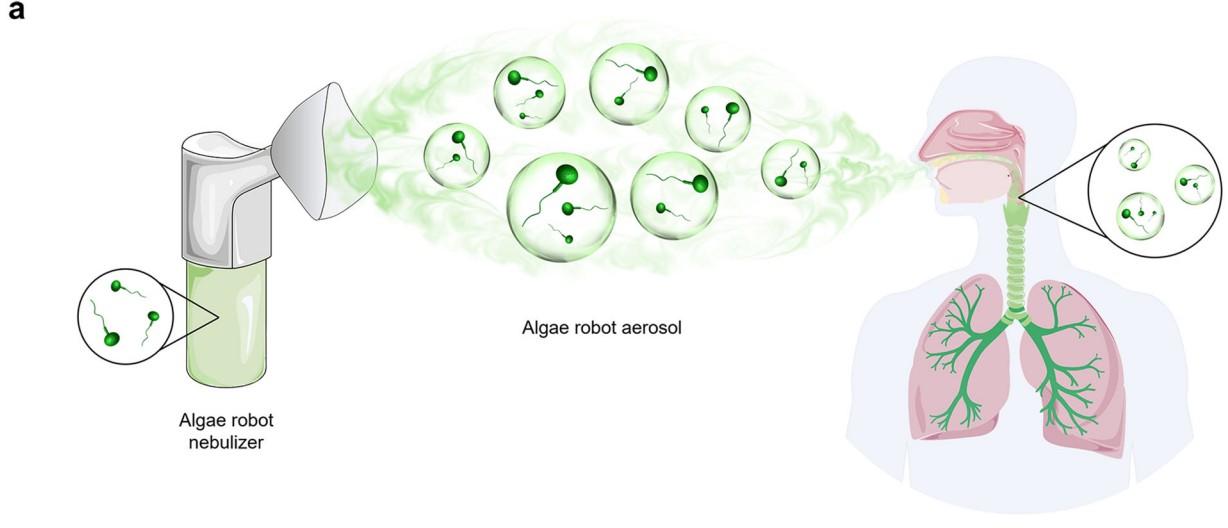

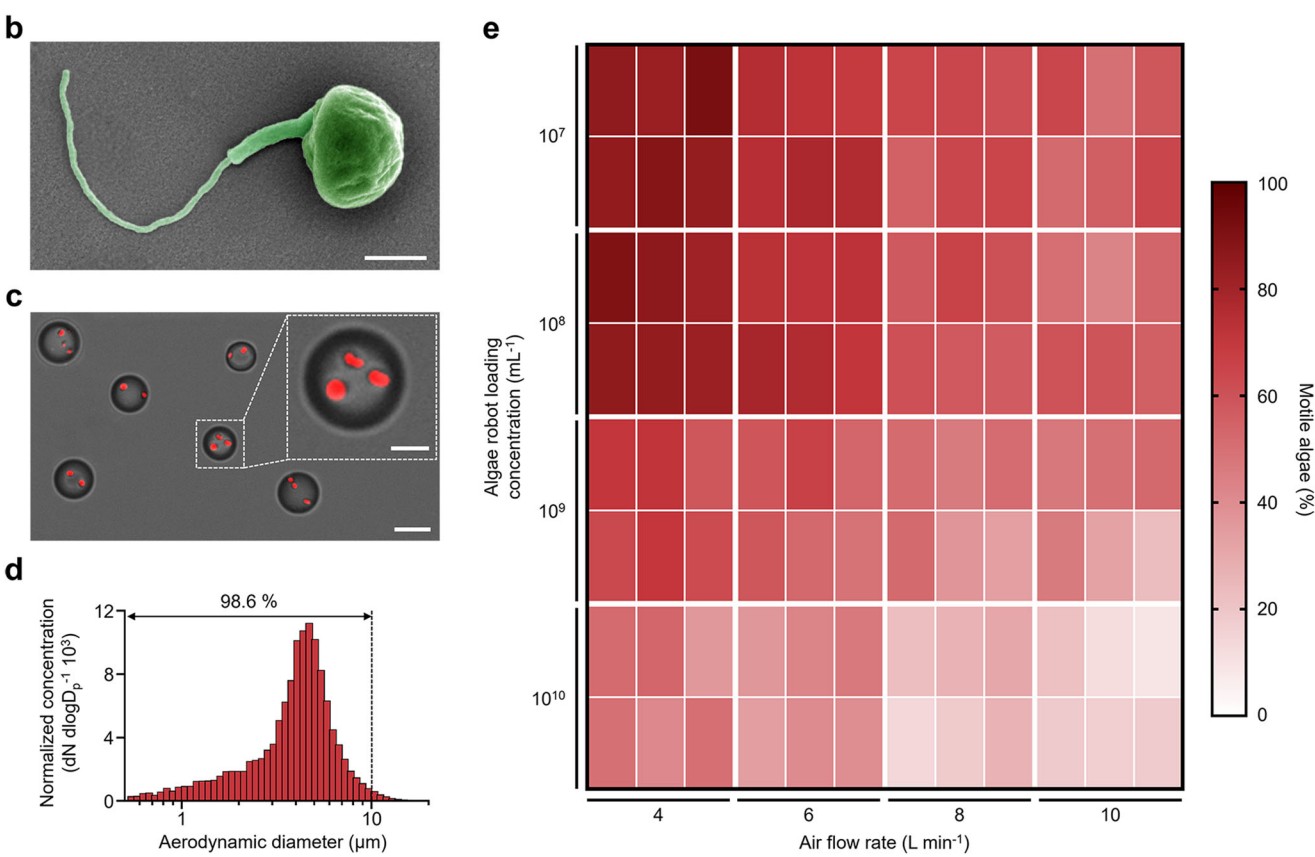

**Fig. 1 | Characterization of algae-based biohybrid microrobot nebulizer platform for inhalable lung delivery. a** A nebulizer is used to aerosolize *M. pusilla* green algae-based biohybrid microrobots. The aerosol droplets are inhaled into the lungs where the biohybrid microrobots actively distribute. **b** Pseudo-colored SEM image of an algae robot. Scale bar, 500 nm. **c** Brightfield (BF) and Cy5 fluorescent channel merged images showing aerosol particles loaded with algae robots (in red).

Scale bars, 5 μm (main) and 2 μm (zoomed-in). **d** Representative aerodynamic diameter size distribution of aerosol particles loaded with algae robots. **e** Heat map displaying the motility ratio of algae robots in simulated lung fluid (SLF) after nebulization. The study was conducted at nebulizer air flow rates at 4, 6, 8, and 10 L min$^{-1}$ and green algae robot concentrations of $10^7$, $10^8$, $10^9$, and $10^{10}$ mL$^{-1}$ ($n = 6$). Independent experiments were performed ($n = 3$) for (**b** and **c**) with similar results.

While the above characterizations were conducted at room temperature (22 °C), we further assessed the impact of body temperature (37 °C) on the motion characteristics of algae robots in SLF after a 60 min nebulization (Fig. 2k, l), in preparation for subsequent in vivo studies. The algae robot speed remained stable in SLF over 120 min at 22 °C, ranging from 55.79 ± 7.52 μm s$^{-1}$ at 0 min to 61.37 ± 5.07 μm s$^{-1}$ at

120 min, whereas the speed was reduced to 22.91 ± 5.14 μm s$^{-1}$ after 120 min at 37 °C. Such speed reduction can be attributed to operating the algae in suboptimal culture conditions, leading to a mild suppression of their metabolism. Similar trends were observed for the motility ratio, which remained high at approximately 80% at 22 °C but decreased to around 50 % after 120 min at 37 °C. Analysis of the

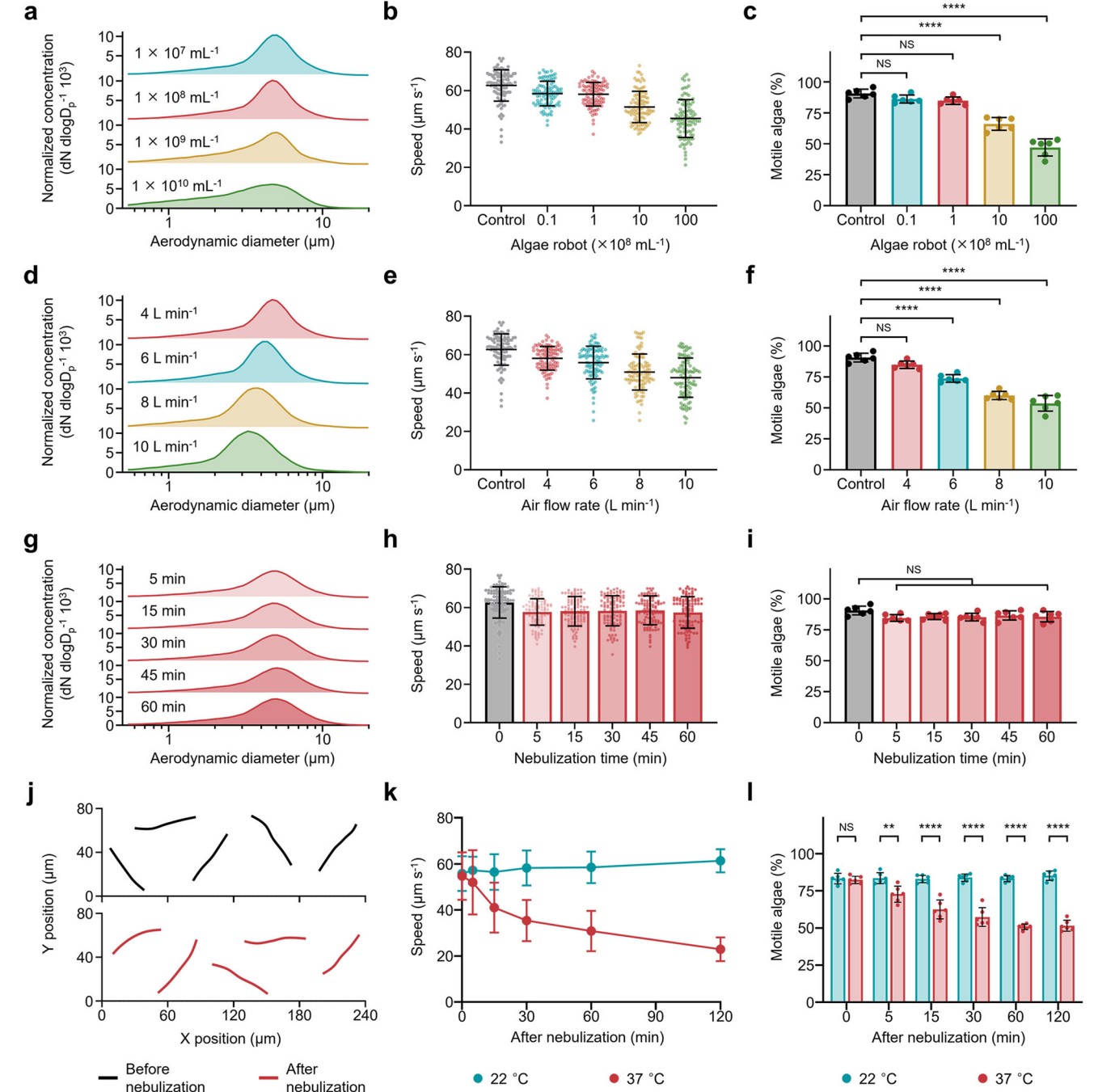

**Fig. 2 | Motility of algae-based biohybrid microrobots after nebulization.**
**a** Aerodynamic size distribution of algae robot aerosol particles at different algae robot loading concentrations of $10^7$, $10^8$, $10^9$, and $10^{10}$ mL$^{-1}$ with a constant nebulizer air flow rate of 4 L min$^{-1}$. **b, c** Algae robot speed (**b**, $n = 100$, n represent the number of individual algae robots, mean ± s.d.) and motile ratio (**c**, $n = 6$, n represent the number of independent experiments, mean ± s.d.) after nebulization in simulated lung fluid (SLF) at room temperature (22 °C). **d** Aerodynamic size distribution of algae robot aerosol particles with different nebulizer air flow rates of 4, 6, 8, and 10 L min$^{-1}$ at an algae robot loading concentration of $10^8$ mL$^{-1}$. **e, f** Comparison of algae robot motion speed (**e**, $n = 100$, n represent the number of individual algae robots, mean ± s.d.) and motile ratio (**f**, $n = 6$, n represent the number of independent experiments, mean ± s.d.) after nebulization in SLF at 22 °C. **g** Aerodynamic size distribution of algae robot aerosol particles at different nebulization times

from 5 to 60 min with a $10^8$ mL$^{-1}$ algae robot concentration and 4 L min$^{-1}$ nebulizer air flow rate. **h, i** Comparison of algae robot motion speed (**h**, $n = 100$, n represent the number of individual algae robots, mean ± s.d.) and motile ratio (**i**, $n = 6$, n represent the number of independent experiments, mean ± s.d.) after nebulization in SLF at 22 °C. **j** Representative trajectories of algae robots (Supplementary Movies 3) corresponding to 1 s of motion before (top) and after (bottom) nebulization ($n = 5$). **k, l** Comparison of algae robot motion speed (**k**, $n = 100$, n represents the number of individual algae robots, mean ± s.d.) and motile ratio (**l**, $n = 6$, n represent the number of independent experiments, mean ± s.d.) after 60 min of nebulization in SLF at 22 °C and body temperature (37 °C). Statistical analysis for the motile algae ratio was performed using repeated-measure one-way analysis of variance (ANOVA). NS: $P > 0.05$, **$P \le 0.01$, ****$P \le 0.0001$.

trajectories of the algae robots at 37 °C also indicated that there was a decrease in speed over time (Supplementary Fig. 13 and Supplementary Movie 6). To further evaluate the motion capability in pathologically relevant simulated lung environments, we measured the speed of

the algae robots in the presence of mucus and cytokines at pH = 6 at 37 °C. It was observed that the speed of algae robot decreased from 59.47 ± 6.72 µm s$^{-1}$ in normal conditions to 44.84 ± 10.01 µm s$^{-1}$ in pathological conditions at 0 h and further decreased from

$25.85 \pm 6.99 \, \mu m \, s^{-1}$ in normal conditions to $15.91 \pm 7.83 \, \mu m \, s^{-1}$ in pathological conditions at 2 h. However, negligible impact was observed on the motility ratio and viability of the algae robots at the pathological conditions (Supplementary Fig. 14). Overall, these studies demonstrated that, with the optimal parameters selected, the nebulized algae robots could retain a significant portion of their motion characteristics, even at body temperature.

## In vivo distribution, retention, and clearance in the lungs

To investigate the lung distribution of algae robots delivered by nebulization, we first established a chamber-based system that did not require mice to be put under anesthesia (Fig. 3a). Prior to in vivo study, we confirmed that the algae robots exhibited negligible cytotoxicity in vitro (Supplementary Fig. 15). We then assessed the time-dependent inhalation profile to determine delivery efficiency in vivo. During the 60-minute nebulization period, the mice remained active and exhibited no signs of respiratory distress or discomfort, supporting the user-friendly nature of this non-invasive delivery method (Supplementary Movie 7). At 5, 15, 30, 45, and 60 min after the start of administration, groups of mice were euthanized, and their lungs were imaged ex vivo to detect algae autofluorescence (Fig. 3b). It was observed that the algae robots gradually and uniformly distributed throughout the entire lungs. Deflagellated algae incapable of motion (denoted as 'static algae') were used as a control group and characterized using SEM (Supplementary Fig. 16). Notably, the autofluorescence signal measured in lungs administered with the same dosage of static algae and algae robots were near identical after up to 60 min of nebulization (Fig. 3c and Supplementary Fig. 17).

Subsequently, we studied the long-term retention profile in the lungs following 60 min of administration via nebulization between the algae robot and static algae groups; algae robots administered via intratracheal instillation were added as a control group. Analysis of the autofluorescence in the lungs revealed that the algae robot group exhibited a negligible decrease within 24 h, and the signal remained detectable up to 120 h (Fig. 3d, e). In contrast, the static algae group showed a significant reduction in signal within 8 h, with a return back to near-baseline levels by 24 h. The same trends were observed when analyzing lung homogenate samples, where a gradual decline in fluorescent signal was observed for the algae robots over a duration of 168 h (Fig. 3f and Supplementary Fig. 18). For the static algae group, the signal was reduced by nearly 90% within the first 24 h. These results suggested that the active motility of the algae robots significantly improved their retention time in the lung post nebulization. Furthermore, we compared the retention profiles of algae robots administered by nebulization versus intratracheal instillation, a method that delivers algae robots with minimal effect on motility, and we found no significant differences in signal within 24 h at an equivalent dosage (Fig. 3d–f and Supplementary Fig. 18). These results were consistent with an in vitro experiment, demonstrating that algae robots retained their ability to evade macrophage uptake post nebulization (Supplementary Fig. 19).

Regarding the clearance mechanism of algae from the lungs, we hypothesized that uptake by alveolar macrophages was a key factor. First, it was confirmed that the autofluorescence intensity of the algae robots and static algae in SLF at body temperature remained stable for up to 72 h in vitro without light exposure (Supplementary Fig. 20). Subsequently, alveolar macrophages were harvested from the lungs at various time points following 60 min of nebulization with either algae robots or static algae, and uptake was analyzed using flow cytometry (Fig. 3g, h and Supplementary Fig. 21). For the first 72 h, the algae robots showed negligible uptake by alveolar macrophages, which significantly increased by 120 h. Conversely, static algae uptake was observed in ~30% of macrophages within 4 h, peaking at 8 h post nebulization before decreasing at later time points. The observed decrease in signal can be attributed to the degradation of the algae

after uptake, which leads to the loss of their autofluorescence. These findings indicate that macrophage phagocytosis is a critical factor in the observed algae clearance kinetics within the lungs.

## In vivo antibacterial efficacy of nebulized algae robot carrying an antibiotic payload

Having demonstrated the enhanced retention of inhaled algae robots in the lungs, we proceeded to evaluate the utility of the platform to treat a murine model of MRSA-induced acute pneumonia. Vancomycin (Vanc), a widely used antibiotic for treating MRSA infections[40], was encapsulated within platelet membrane-coated PLGA nanoparticles (denoted as 'PNP(Vanc)'). The core-shell structure of these nanoparticles was visualized through transmission electron microscopy (TEM), and successful association of the platelet membrane with the PLGA cores was confirmed by fluorescent imaging (Supplementary Fig. 22a, b). Post membrane coating, dynamic light scattering (DLS) measurements revealed an increase in hydrodynamic size from 89 nm to 110 nm, accompanied by a shift in surface zeta potential from $-41$ mV to $-30$ mV (Supplementary Fig. 22c, d). Azido-PEG$_4$-$N$-hydroxysuccinimidyl (NHS) ester-modified PNP(Vanc) were then conjugated to the surface of dibenzocyclooctyne (DBCO)-PEG$_4$-NHS ester-modified algae robots via click chemistry (Supplementary Fig. 23). The resulting drug-carrying biohybrid microrobots (denoted as 'algae-PNP(Vanc)-robot') were characterized by SEM to visualize the successful functionalization of the algae robots with PNP(Vanc) (Fig. 4a). Subsequently, we assessed the Vanc loading onto the algae robot, and it was observed that the amount of loaded drug increased proportionally with the increase of algae number (Fig. 4b). For further studies, we selected the algae-PNP(Vanc)-robot formulation consisting of 10 μg Vanc loaded onto $1 \times 10^8$ algae, which ensured peak motility in SLF (Supplementary Fig. 24). The release pattern of the algae-PNP(Vanc)-robot was similar to that of PNP(Vanc), with ~50% of the Vanc being released within the first 24 h, followed by a gradual release reaching up to ~90% by 72 h (Supplementary Fig. 25). Subsequently, the in vitro study of minimal inhibitory concentration (MIC) against MRSA was assessed for the algae-PNP(Vanc)-robot alongside different control groups, including free Vanc, PNP(Vanc), and static algae functionalized with PNP(Vanc) (denoted as 'static algae-PNP(Vanc)'). The static algae-PNP(Vanc) was characterized by SEM imaging, which revealed the absence of the flagella required for motility (Supplementary Fig. 26). It was revealed that the bacterial growth was inhibited at a Vanc concentration of $2 \, \mu g \, mL^{-1}$ across all groups (Supplementary Fig. 27), which was consistent with previous reports[41]. Further enumeration of bacterial colony-forming units (CFU) demonstrated that $2 \, \mu g \, mL^{-1}$ Vanc was also the minimal bactericidal concentration (MBC) against MRSA in all groups (Supplementary Fig. 28).

To evaluate the feasibility of delivering algae-PNP(Vanc)-robot via nebulization, we utilized optimized nebulizer parameters with a loading of $1 \times 10^8 \, mL^{-1}$ of algae, corresponding to $10 \, \mu g \, mL^{-1}$ of Vanc, along with a system air flow rate of $4 \, L \, min^{-1}$. The aerodynamic size distribution of the nebulized droplets for algae-PNP(Vanc)-robot, static algae-PNP(Vanc), and PNP(Vanc) were all similar to that of a $1 \times$ PBS buffer control group (Supplementary Fig. 29). Flow cytometry analysis confirmed that the nebulization did not affect the binding of PNP(Vanc) to the algae robots, with levels remaining consistent before and after nebulization (Supplementary Fig. 30). The post-nebulization motility profiles of the algae-PNP(Vanc)-robot were analyzed in SLF at 37 °C without light. This study revealed that, while their speed and motile ratio declined after 2 h, nebulization had minimal impact on these parameters, and the viability of the algae remained consistent at around 95% (Supplementary Fig. 31 and Supplementary Movie 8). To investigate the delivery of the drug payload via inhalation, we quantified the total Vanc per lung after up to 60 min of administration via nebulization. The findings revealed that ~ 8.5 μg could be delivered per lung after 60 min of nebulization with algae-PNP(Vanc)-robot (Fig. 4c).

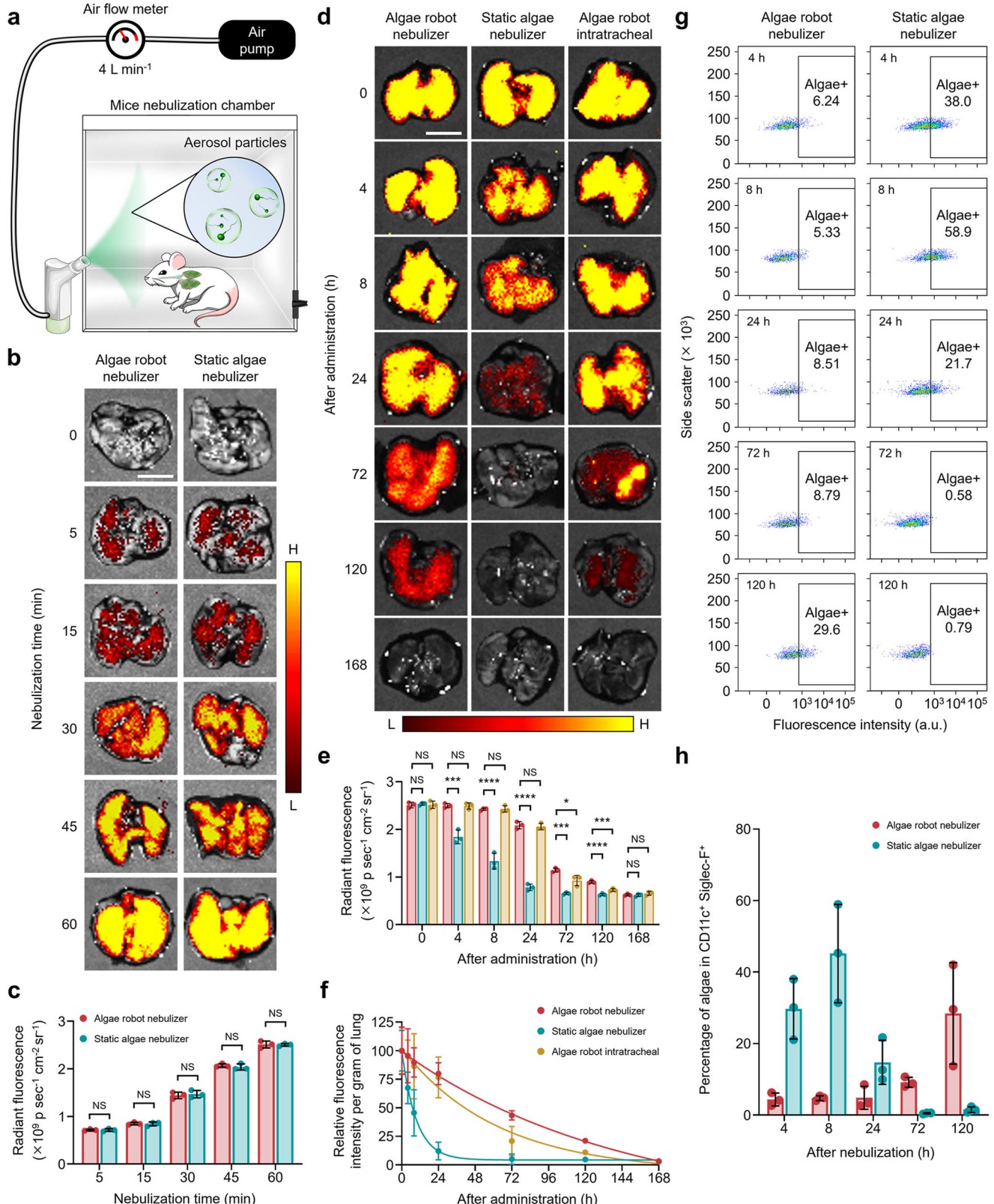

Comparable results were obtained for the static algae-PNP(Vanc) and PNP(Vanc) groups (Supplementary Fig. 32).

The treatment efficacy of algae-PNP(Vanc)-robot delivered via nebulization was evaluated using a murine model of MRSA-induced acute pneumonia. Lung infections were established by inoculating MRSA into the mice lungs via intratracheal administration. Following a 1-h recovery period, the mice received a 60-min nebulization treatment with algae-PNP(Vanc)-robot or with various control groups (PNP(Vanc), static algae-PNP(Vanc), and 1 × PBS buffer). As an additional control based on the current clinical standard, intravenous (IV) administration of free Vanc was selected. At 24 h after administration, the lungs were collected, homogenized, and analyzed for bacterial counts (Fig. 4d). The bacterial burden in lungs treated with algae-PNP(Vanc)-robot via nebulization was $1.95 \times 10^4$ CFU g$^{-1}$, reflecting a

**Fig. 3 | Inhalation and lung distribution of algae-based biohybrid microrobots.**
**a** Schematic of the mice nebulization system. **b** Representative ex vivo fluorescent images of the lungs after different nebulization administration times with algae robots or static algae. H, high signal; L, low signal. Scale bars, 5 mm. **c** Quantification of fluorescence for the lung samples in (**b**) ($n = 3$, n represents the number of individual mice, mean ± s.d.). **d** Representative ex vivo fluorescent images of the lungs at various time points after 60 min of nebulization administration with algae robot or static algae, or intratracheal administration with algae robot. H, high signal; L, low signal. Scale bars, 5 mm. **e**, **f** Total (e) and normalized (f) fluorescence quantification of the lung samples in (**d**) ($n = 3$, n represent the number of individual

mice, mean ± s.d.). **g** Representative flow cytometry dot plots of algae robot and static algae uptake by alveolar macrophages (CD11c$^+$Siglec-F$^+$) in vivo at different time points after 60 min of nebulization administration. **h** Quantification of algae robot and static algae uptake in (**g**) ($n = 3$, n represent the number of individual mice, mean ± s.d.). Statistical analysis was performed using an unpaired two-tailed $t$ test for (**c**) and using repeated-measure one-way analysis of variance (ANOVA) for (**e**). NS: $P > 0.05$, *$P \le 0.05$, ***$P \le 0.001$, ****$P \le 0.0001$. The nonlinear fit curve for (**f**) was performed using a nonlinear regression (curve fit) model through GraphPad Prism 10.

---

reduction of four orders of magnitude compared to the 1 × PBS control group ($1.07 \times 10^8$ CFU g$^{-1}$). Algae-PNP(Vanc)-robot treatment also resulted in a significantly lower bacterial load compared to the static algae-PNP(Vanc) group ($6.21 \times 10^6$ CFU g$^{-1}$) and the PNP(Vanc) group ($8.28 \times 10^6$ CFU g$^{-1}$). Moreover, nebulization of the algae-PNP(Vanc)-robot resulted in markedly higher efficacy than an equivalent IV dose of free Vanc (8.5 μg per mouse, one dose) ($8.30 \times 10^7$ CFU g$^{-1}$) and showed comparable efficacy to clinical IV doses of free Vanc (30 mg kg$^{-1}$, equivalent to 600 μg per mouse, two doses) ($6.15 \times 10^4$ CFU g$^{-1}$). In a separate survival study, infected mice treated with algae-PNP(Vanc)-robot via nebulization had a 100% survival rate over the 60-day study period (Fig. 4e). In contrast, mice administered with 1 × PBS buffer, static algae-PNP(Vanc), or an equivalent IV dose of free Vanc all died within 3 days. Long-term survival rates for mice treated with PNP(Vanc) and clinical IV doses of free Vanc were 16.67% and 66.67%, respectively.

### In vivo biosafety evaluation of algae-PNP(Vanc)-robot via nebulization
Finally, we evaluated the biosafety of algae-PNP(Vanc)-robot on 1, 7, and 14 days post nebulization. The comprehensive analysis of blood chemistry and major blood cell populations indicated that mice administered with algae-PNP(Vanc)-robot via nebulization maintained all values within the normal healthy range, demonstrating no adverse effects (Fig. 4f, g). To assess potential tissue-level toxicity, H&E staining was conducted on the heart, liver, spleen, lungs, and kidneys, revealing no abnormalities at either short or long intervals (Fig. 4h). To evaluate the possibility of chronic inflammatory responses, pro-inflammatory cytokine (TNF-α, IL-6, and IL-1β) levels from blood samples on days 1, 7 and 14 post nebulization were quantified, indicating no systemic immune reaction due to the nebulization administration of algae-PNP(Vanc)-robot. (Supplementary Fig. 33). These findings indicated the favorable biosafety profile of the algae-PNP(Vanc)-robot formulation and administered noninvasively by nebulization.

### Discussion
In this work, we demonstrated the first biohybrid microrobot nebulizer system designed for therapeutic pulmonary delivery by inhalation. In addition to the non-invasive nature and enhanced patient comfort of administration by nebulization, the aerosol particles generated by this approach enable the loaded biohybrid microrobots to effectively reach the lungs. The picoeukaryote algae microrobots were encapsulated efficiently within the small-sized aerosol particles and maintained their motility after inhalation delivery. Unlike static drug carriers, which frequently encounter limitations such as rapid clearance by alveolar macrophages[42,43], the robust self-propulsion capabilities of the algae robots significantly inhibited macrophage phagocytosis. This led to enhanced retention in mouse lungs over an extended period of several days. Through pulmonary inhalation delivery of the drug-carrying algae robots, we observed significant enhancements in bactericidal efficacy and survival rates in a murine model of acute pneumonia compared to static algae and nanoparticle only controls. Our results confirmed that the inherent long-lasting self-propulsion of the biohybrid microrobots makes them an attractive choice for delivering

therapeutic agents to the lungs. An evaluation of the biosafety of the algae-based biohybrid microrobot nebulizer system illustrated that our formulation had negligible impact on blood chemistry, blood cell counts, and organ integrity, underscoring its potential suitability towards clinical transition.

The aforementioned results achieved using our algae-based biohybrid microrobot nebulizer system underscore its considerable promise for lung treatment amidst the growing prevalence of respiratory diseases. In comparison to intratracheal administration, the nebulization route offers the major advantage of being non-invasive, providing a more comfortable and patient-friendly option. The algae-based biohybrid microrobot effectively maintains its self-propulsion ability post nebulization, resulting in uniform distribution and prolonged retention in the lungs, thereby enhancing therapeutic efficiency and reducing potential side effects. Therefore, besides MRSA-induced acute pneumonia, our system could potentially be employed for the treatment of conditions such as asthma, tuberculosis, acute lung injury, or chronic obstructive pulmonary disease[44–46]. The prolonged lung retention ensures that therapeutic agents remain in the lungs for extended periods, leading to enhanced disease management and improved patient outcomes while reducing the frequency of administration to improve compliance. Moving toward clinical transition, studies will need to be conducted in larger animal models to validate safety and therapeutic efficacy when delivering the aerosols via a mouthpiece or mask versus the chamber-based approach used in the present work. Scaling this administration method for human use will require researchers to account for the differences in lung structure and size, towards maintaining the efficient distribution and retention we achieved here in murine models. Factors such as aerosol particle size, flow rate, and total inhaled dose will need to be optimized for human lungs. Our active biohybrid delivery system can be adapted for use with various payloads, and properties such as drug release rate can be tailored to meet specific clinical needs. Overall, microrobots offer distinctive functionalities that enable them to outperform common passive delivery systems. With aerosol inhalation delivery, the scalability of this microrobot nebulizer platform makes it a compelling candidate for future clinical translation of microrobot technology, offering broad applications in managing various lung indications.

## Methods
### Green algae culture
*Micromonas pusilla* (strain CCMP 1545), a eukaryotic and photosynthetic microorganism sourced from the National Center for Marine Algae and Microbiota, was cultivated in L1-Si medium (Bigelow). The cultures were maintained at a room temperature of approximately 22 °C under a controlled photoperiod of 14 h of light followed by 10 h of darkness.

### Characterization of nebulizer system
The algae-based biohybrid microrobot nebulizer system, displayed in Supplementary Fig. 1, consisted of an air pump (Trek S, PARI), an airflow controller (SCF 06, Tailonz), an airflow meter (LZQ-7, Hilitand), and an algae robot nebulizer cup (Trek S, PARI), interconnected via

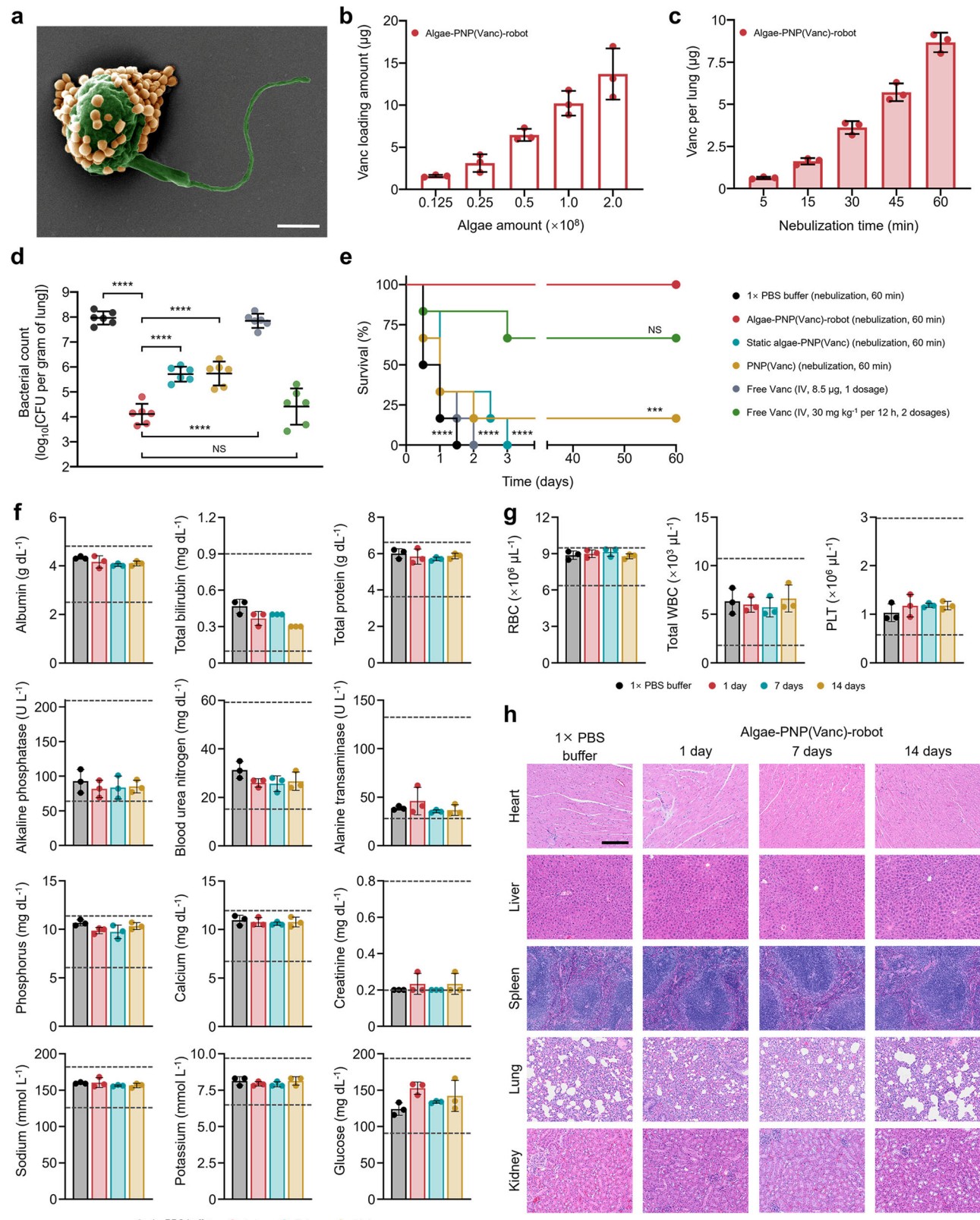

wing tip tubes (Trek S, PARI). The air pump generated a consistent airflow throughout the system, facilitated by the wing tip tubes. The airflow controller precisely regulated the air flow rate, while the airflow meter provided real-time monitoring to ensure accuracy. The nebulizer cup was utilized to hold aqueous samples, with a capacity ranging from a minimum of 2 mL to a maximum of 8 mL. Upon securely connecting all components of the nebulizer system, activating the air

pump produced a stable aerosol flow through the nebulizer nozzle, ensuring efficient sample nebulization.

## Characterization of aerosol particles

The aerosol particle size was assessed using an aerodynamic particle sizer (TSI 3320), with measurements standardized following the TSI 'Aerosol Statistics Lognormal Distributions and dN/dlogD$_p$' protocol.

**Fig. 4 | In vivo therapeutic efficacy and safety evaluation of drug-loaded algae robots administered via nebulization. a** Pseudo-colored SEM image of an algae robot functionalized with platelet membrane-coated vancomycin-loaded nanoparticles (denoted as 'algae-PNP(Vanc)-robot'). Scale bar, 500 nm. **b** Quantification of Vanc loading amount at different algae robot input amounts ($n = 3$, mean ± s.d.). **c** Quantification of Vanc inhalation amount per lung after different nebulization times with algae-PNP(Vanc)-robot ($n = 3$, mean ± s.d.). **d** Bacterial load in the lungs of MRSA-infected mice after treatment with 1 × PBS buffer, algae-PNP(Vanc)-robot, static algae-PNP(Vanc), and PNP(Vanc) via 60 min of nebulization, as well as free Vanc at one dosage of 8.5 μg and two clinical dosages of 30 mg kg⁻¹ via IV administration ($n = 6$, n represent the number of individual mice, geometric mean ± s.d.). **e** Survival of MRSA-infected mice after treatment with 1 × PBS buffer, algae-PNP(Vanc)-robot, static algae-PNP(Vanc), and PNP(Vanc) via 60 min of nebulization, as well as free Vanc at one dosage of 8.5 μg and two clinical dosages of

30 mg kg⁻¹ via IV administration ($n = 6$, n represent the number of individual mice). **f** Comprehensive blood chemistry panels conducted 1 day after 60 min of nebulization administration with 1 × PBS buffer or 1, 7, and 14 days after 60 min of nebulization administration with algae-PNP(Vanc)-robot ($n = 3$, mean ± s.d.). **g** Counts of various blood cells conducted 1 day after 60 min of nebulization administration with 1 × PBS buffer or 1, 7, and 14 days after 60 min of nebulization administration with algae-PNP(Vanc)-robot ($n = 3$, mean ± s.d.). **h** Haematoxylin and eosin staining of histology sections from major organs 1, 7, and 14 days after 60 min of nebulization administration with 1 × PBS buffer or algae-PNP(Vanc)-robot. Scale bar, 200 μm. Independent experiments were performed ($n = 3$) with similar results. Statistical analysis was performed using repeated-measure one-way analysis of variance (ANOVA) for (**d**) and using the log-rank (Mantel-Cox) test for (**e**). NS: $P > 0.05$, ***$P \le 0.001$, ****$P \le 0.0001$.

Specifically, 'dN' is the total aerosol particle concentration (particle cm⁻³), and 'dlogD$_p$' is the difference in the log of the machine measurement channel width (51 channels in total). The aerosol particle concentration is divided by the channel width, giving a normalized concentration value that is independent of the channel width. For the loading concentration study of algae robots, 8 mL samples at varying concentrations ($1 \times 10^7$ mL⁻¹, $1 \times 10^8$ mL⁻¹, $1 \times 10^9$ mL⁻¹, and $1 \times 10^{10}$ mL⁻¹) were loaded into the nebulizer reservoir. Maintaining a constant air flow rate of 4 L min⁻¹, aerosol particle sizes were continuously measured, integrating 1 min intervals to ensure the stability and accuracy of the measurements, which were then analyzed and standardized. In the air flow rate study, the system air flow rate was adjusted to 4, 6, 8, and 10 L min⁻¹ with an algae robot loading of $1 \times 10^8$ mL⁻¹. Aerosol particle sizes were measured continuously, integrating 1-min results for stability, followed by analysis and standardization. To evaluate the stability of the nebulizer system at an algae robot loading of $1 \times 10^8$ mL⁻¹ and a system air flow rate of 4 L min⁻¹, aerosol particle sizes were measured after 5, 15, 30, 45, and 60 min of nebulization, with data subsequently analyzed and standardized. For visualization, the aerosol particles were collected into olive oil (Pompeian) and then observed using a fluorescence microscope (EVOS FL) in both the brightfield and Cy5 fluorescent channels. This protocol was also applied to assess the particle sizes of aerosols containing 1 × PBS buffer, algae-PNP(Vanc)-robot, static algae-PNP(Vanc), and PNP(Vanc).

## Human platelet isolation and membrane derivation

Human type O⁻ blood, anticoagulated with 1.5 mg mL⁻¹ ethylenediaminetetraacetic acid (EDTA, Thermo Fisher Scientific) was sourced from BioreclamationIVT and processed for platelet collection approximately 16 h post blood draw. To isolate platelets, blood samples were first adjusted to a final EDTA concentration of 5 mM, followed by centrifugation at $100 \times g$ for 20 min at 22 °C, effectively separating red and white blood cells. The resultant platelet-rich plasma (PRP) underwent a subsequent centrifugation at $100 \times g$ for an additional 20 min to ensure the removal of any remaining blood cells. To inhibit platelet activation, PBS buffer containing 1 mM EDTA and 2 μM prostaglandin E1 (PGE1, Sigma-Aldrich) was introduced to the purified PRP. Platelets were then pelleted by centrifugation at $800 \times g$ for 20 min at 22 °C. The supernatant was carefully discarded, and the platelets were resuspended in 1 × PBS containing 1 mM EDTA, supplemented with protease inhibitor tablets (Pierce). The method was described in a previously published protocol[35].

Platelet membranes were derived from a suspension of $2 \times 10^9$ platelets mL⁻¹ through a series of repeated freeze-thaw cycles[47]. Platelet aliquots were initially frozen at − 80 °C, thawed at 22 °C, and centrifuged at $4000 \times g$ for 3 min. This process was repeated 3 times, with thorough washes in 1 × PBS buffer supplemented with protease inhibitor tablets between cycles. The resulting platelet membrane pellets were resuspended in ultrapure water (Thermo Fisher Scientific) and stored at − 80 °C for subsequent analyses. The membrane protein

concentration was determined using the bicinchoninic acid (BCA) assay (Life Technologies).

## Fabrication of PNP(Vanc)

To synthesize drug-loaded polymeric cores, a reported method with slight modifications was used[48]. First, 50 μL of 200 mg mL⁻¹ vancomycin hydrochloride (Vanc, Santa Cruz Biotechnology) in ultrapure water was emulsified into 500 μL of a 40 mg mL⁻¹ 50:50 PLGA (0.67 dl g⁻¹, Lactel Absorbable Polymers) solution in dichloromethane (DCM) using an ultrasonic probe sonicator (Thermo Fisher Scientific) at 10 W power. The sonication process, performed within an ice bath, consisted of 2 min of alternating 2-s power on and off cycles. The emulsion was then transferred to 5 mL of aqueous solution and was further sonicated for 3 min. To ensure complete DCM evaporation, the emulsion was stirred for 4 h at 22 °C. For fluorescence labeling, 4 μL of DiO (excitation wavelength ($\lambda_{ex}$)/emission wavelength ($\lambda_{em}$) = 484/501 nm, Thermo Fisher Scientific) was added to 1 mL of DCM containing 20 mg of PLGA. The synthesized Vanc-loaded PLGA cores were then mixed with platelet membranes at a polymer-to-protein weight ratio of 1:1. This mixture underwent 3 min of sonication using a bath sonicator (Fisherbrand 11201 series, Thermo Fisher Scientific). Finally, the sample was washed 5 times with ultrapure water via centrifugation at $16,100 \times g$ for 5 min each, and then suspended in ultrapure water for future use.

## Characterization of PNP(Vanc)

The morphology of the nanoparticles was visualized using a transmission electron microscope (FEI 200 kV Sphera) following staining with 0.2 wt% uranyl acetate (Electron Microscopy Sciences). For fluorescence microscopy analysis (EVOS FL), PNP(Vanc) was observed to identify DiO-loaded PLGA cores via the green fluorescent protein (GFP) channel and DiI ($\lambda_{ex}/\lambda_{em}$ = 550/564 nm)-labeled platelet membranes via the red fluorescent protein (RFP) channel. Furthermore, the hydrodynamic size and zeta potential of the nanoparticles were assessed using dynamic light scattering (ZEN 3600 Zetasizer, Malvern).

## Modification of algae-PNP(Vanc)-robot

To prepare the algae-PNP(Vanc)-robot, the algae robots were washed 5 times in 1 × PBS buffer with centrifugation at $1500 \times g$ for 3 min and the concentration was adjusted to $1 \times 10^8$ mL⁻¹. Subsequently, 1 mL of algae robots was mixed with 2 μL of 20 mM DBCO-PEG₄-NHS ester (Click Chemistry Tools) and incubated at 22 °C for 1 h. Simultaneously, 1 mL of PNP(Vanc) at 1 mg mL⁻¹ in water was mixed with 4 μL of 10 mM azido-PEG₄-NHS ester (Click Chemistry Tools) and vortex-incubated at 22 °C for 1 h at 700 rpm. Post incubation, both the DBCO-labeled algae robots and azido-labeled PNP(Vanc) underwent additional washing in 1 × PBS buffer. Algae robots were centrifuged at $1500 \times g$ for 2 min, and PNP(Vanc) at $16,100 \times g$ for 3 min, with each washing step repeated 5 times to ensure the removal of excess reagents. Conjugation was

achieved by mixing the DBCO-labeled algae robots with the azido-labeled PNP(Vanc) and incubating for 1 h at 22 °C. After conjugation, the resultant algae-PNP(Vanc)-robots were washed twice with 1 × PBS buffer by centrifugation at $1500 \times g$ for 2 min and stored in 1 × PBS buffer for further characterization. For the preparation of static algae-PNP(Vanc), after incubation of algae robots with 250 mM acetic acid, the resulting deflagellated green algae were subjected to a similar procedure for PNP(Vanc) conjugation.

## Characterization of algae-PNP(Vanc)-robot
Samples of algae-PNP(Vanc)-robots were fixed using 5% glutaraldehyde (Sigma-Aldrich) at a 1:1 volume ratio and then stored at 4 °C overnight. After fixation, the samples were subjected to 3 washes with ultrapure water by centrifugation at $800 \times g$ for 2 min. The samples were then dried overnight. Following the drying process, the samples were sputter-coated with palladium. The algae-PNP(Vanc)-robots were then characterized using a Zeiss Sigma 500 SEM instrument at an acceleration voltage of 3 kV. Static algae-PNP(Vanc), algae robots, and static algae were prepared and analyzed using the same protocol.

## Motility analysis
The motility of the algae robot before and after nebulization was assessed in simulated lung fluid (SLF, Supplementary Table 1) at both 22 °C and 37 °C. Various loadings ($1 \times 10^7$ mL$^{-1}$, $1 \times 10^8$ mL$^{-1}$, $1 \times 10^9$ mL$^{-1}$, and $1 \times 10^{10}$ mL$^{-1}$) of algae robot in 1 × PBS buffer were aerosolized at different system air flow rates (4, 6, 8, and 10 L min$^{-1}$). The resulting algae robot-containing aerosol particles were then collected in SLF at 22 °C. To evaluate post-nebulization motility, the motility of the algae robot was also studied in a simulated solution containing lung mucus and cytokines at pH = 6 at 37 °C. The corresponding motion changes were measured at 0, 5, 15, 30, 60, and 120 min. Videos of the motion in both the brightfield and Cy5 fluorescent channels were recorded using an optical microscope (Nikon ECLIPSE Ti-S/L100) equipped with a digital camera (Hamamatsu, C11440) under 20× or 40× objective lenses (Nikon). The NIS-Elements tracking module (Nikon) was utilized to measure the corresponding algae robot speed in SLF. The motility ratio of the algae robot was determined by analyzing random 2 s segments from the recorded videos under various conditions. The motility characteristics of the algae-PNP(Vanc)-robot were measured following the same procedures.

## Algae viability evaluation
To evaluate algae viability, algae under various conditions were stained with 5 μM SYTOX GFP (Thermo Fisher Scientific). Fluorescence microscopy (EVOS FL) was employed to visualize the bright green fluorescence in the GFP channel emitted by dead algae cells. According to established protocols[49], the fluorescence intensity of SYTOX ($\lambda_{ex}/\lambda_{em} = 504/523$ nm) was quantified using a plate reader. Algae viability was subsequently determined by calculating the ratio of live to dead cells, observed under an Invitrogen EVOS FL fluorescence microscope, leveraging the SYTOX GFP signal.

## In vitro cytotoxicity evaluation
To evaluate cytotoxicity, NL-20 cells (ATCC CRL-2503) and J774A.1 cells (ATCC TIB-67) were plated in 96-well plates at a density of $5 \times 10^4$ cells per well. The cells were exposed to algae robot at various algae-to-cell ratios (0.25, 0.5, 1, 2, 4, 8, 16, 32, 64, 128, 256, and 512) for a 24 hour period at 37 °C. The cell viability was assessed using the Cell-Titer Aqueous One Solution Cell Proliferation Assay (MTS, Promega) according to the manufacturer's guidelines.

## Animal care
Seven-week-old male CD-1 mice, obtained from Charles River Laboratories, were housed in the animal facility at the University of California

San Diego (UCSD), in compliance with federal, state, local, and National Institutes of Health (NIH) regulations. The mice were maintained under optimal conditions, including a 12-hour light/dark cycle, controlled temperature, and regulated humidity. All experimental procedures were conducted in strict adherence to NIH guidelines for pain mitigation and humane euthanasia. These protocols received approval from the Institutional Animal Care and Use Committee (IACUC) at UCSD.

## Ethics
Every experiment involving animals, human participants, or clinical samples has been carried out following a protocol approved by an ethical commission.

## Chamber-based nebulization administration
A medium induction chamber (Thermo Fisher Scientific) was employed for nebulization administration, adhering to the manufacturer's guidelines. The nebulizer nozzle was securely connected to the chamber's upper air inlet port using parafilm (Sigma-Aldrich). After assembling the nebulization system and adjusting the airflow rate to 4 L min$^{-1}$, 8 mL of algae robots at $1 \times 10^8$ mL$^{-1}$ in 1 × PBS buffer were loaded into the nebulizer reservoir. Mice (3 per chamber) were allowed a 5-minute acclimation period before initiating the nebulization. The air pump was then activated to begin the nebulization process. Throughout the administration, the status of the mice was closely monitored to ensure the procedure's success. After a 60-minute nebulization session, the air pump was deactivated, and the mice were returned to their cages. The chamber was subsequently washed with tap water in preparation for the next administration. This protocol was consistently applied for nebulization treatments using 1 × PBS buffer, static algae, algae-PNP(Vanc)-robot, static algae-PNP(Vanc), and PNP(Vanc).

## Intratracheal administration
Intratracheal administration was conducted using a protocol adapted from a previous study[50]. Seven-week-old male CD-1 mice (Charles River Laboratories) were anesthetized with a ketamine (100 mg kg$^{-1}$, Pfizer) and xylazine (200 mg kg$^{-1}$, Lloyd Laboratories) cocktail. For the inoculation, 40 μl of the sample (either algae robot or MRSA in 1 × PBS) were injected into a polytetrafluoroethylene feeding tube (cut to a length of 6 to 8 in) using an insulin needle. The feeding tube was carefully inserted 0.5–1.0 cm into the trachea, and the inoculum was administered directly into the lungs. To ensure optimal retention of the experimental sample, the feeding tube was maintained in position for 30 seconds before removal.

## In vivo inhalation, biodistribution, and retention examination
For the inhalation profile studies, male CD-1 mice were delivered $1 \times 10^8$ mL$^{-1}$ algae robot or static algae via the chamber-based nebulizer system in 1 × PBS buffer at an airflow rate of 4 L min$^{-1}$. The mice were euthanized at 0, 5, 15, 30, 45, and 60 min post treatment, and their lungs were harvested for further analysis. For Vanc dosage profiles, healthy mice were similarly delivered $1 \times 10^8$ mL$^{-1}$ algae-PNP(Vanc)-robot, $1 \times 10^8$ mL$^{-1}$ static algae-PNP(Vanc), or PNP(Vanc), all at an equivalent Vanc concentration of 10 μg mL$^{-1}$, in 1 × PBS buffer at a flow rate of 4 L min$^{-1}$. Following nebulization, the mice were euthanized at the same time intervals, and their lungs were collected and homogenized using a BioSpec Mini-BeadBeater-16. Vanc concentrations were quantified by liquid chromatography-mass spectrometry at the Molecular Mass Spectrometry Facility at UCSD. For biodistribution and retention studies, male CD-1 mice underwent a 60-minute nebulization with 1 × PBS buffer, $1 \times 10^8$ mL$^{-1}$ algae robot, or static algae at an airflow rate of 4 L min$^{-1}$. An additional group of mice received an intratracheal administration of an equivalent amount of algae robots at $8 \times 10^7$ in 40 μl of 1 × PBS buffer. Mice were euthanized at specific time points (0,

4, 8, 24, 72, 120, and 168 h), and their lungs were excised for analysis. Fluorescent ex vivo organ imaging was conducted using a Xenogen IVIS 200 system. The fluorescence of lung homogenate samples was measured with a BioTek Synergy Mx microplate reader.

### In vitro alveolar macrophage phagocytosis

To evaluate in vitro alveolar macrophage phagocytosis, J774A.1 macrophage cells were cultured with Dulbecco's modified Egle medium (Invitrogen) in a 48-well plate at a density of $1 \times 10^6$ cells per well. Pre-nebulized active algae-PNP(Vanc) was adjusted to a concentration of $1 \times 10^8$ per well and incubated at 37 °C. The fluorescence intensity of the mixture was measured at 0, 6, 12, 24, 36. 48, 60, and 72 h using a BioTek Synergy Mx microplate reader. Post-nebulized and static algae-PNP(Vanc) controls were tested with an identical method.

### In vivo alveolar macrophage phagocytosis

To evaluate in vivo alveolar macrophage phagocytosis, male CD-1 mice underwent a 60-minute nebulization administration with either $1 \times 10^8 \, mL^{-1}$ algae robot or static algae at an airflow rate of $4 \, L \, min^{-1}$. Bronchoalveolar lavage fluid (BALF) was collected at 4, 8, 24, 72, and 120 h post nebulization following a modified protocol[11]. Mice were euthanized by $CO_2$ inhalation, and a catheter was inserted into an incision in the exposed trachea. BALF was collected through 3 washes with 0.5 mL of a solution containing 0.5 % (v/v) fetal bovine serum (Gibco) and 2 mM EDTA (Thermo Fisher Scientific) in $1 \times$ PBS buffer, and samples were kept on ice. The BALF was centrifuged at $700 \times g$ for 5 min, and red blood cells were lysed using RBC lysis buffer (BioLegend) according to the manufacturer's instructions. The remaining cells were blocked with 1% fetal bovine serum in $1 \times$ PBS buffer on ice for 30 min and then stained with FITC-conjugated anti-mouse Siglec-F (S17007L, BioLegend) and Pacific Blue conjugated anti-mouse CD11c (N418, BioLegend). After washing to remove any unbound antibodies, the cells were resuspended in $1 \times$ PBS buffer, and data was collected using a Becton Dickinson LSR II flow cytometer. Analysis was performed with FlowJo v10.4 software.

### Vanc loading yield and release analysis

To determine the loading yield of algae-PNP(Vanc)-robots, 1 mL of DBCO-labeled algae robots at various concentrations (0.125, 0.25, 0.5, 1, and $2 \times 10^8 \, mL^{-1}$) was mixed with 1 mg of azido-labeled PNP(Vanc) for 1 h. Free PNP(Vanc) was removed by washing the mixture 3 times at $1500 \times g$ for 3 min. The resulting algae-PNP(Vanc)-robots were suspended in 1 mL of $1 \times$ PBS buffer. After fabrication, the Vanc fluorescence intensity ($\lambda_{ex}/\lambda_{em} = 290/325 \, nm$) of the solution was measured. A standard calibration curve was established using serial dilutions of Vanc solution ($0–32 \, \mu g \, mL^{-1}$) to determine the Vanc concentration in the sample. The release of Vanc was studied by incubating 1 mL of algae-PNP(Vanc)-robots in SLF at 37 °C over 96 h, with fluorescence measurements used to monitor the concentration.

### Evaluation of bactericidal efficiency in vitro

The MRSA strain USA300 (ATCC, BAA-1717) was initially cultured in tryptic soy broth (TSB, Sigma-Aldrich) at 37 °C. To determine the minimal inhibitory concentration (MIC) of Vanc, a suspension of MRSA at $1 \times 10^7 \, CFU \, mL^{-1}$ was mixed with algae-PNP(Vanc)-robot at Vanc concentrations of 0.5, 1, 2, 4, 6, and $8 \, \mu g \, mL^{-1}$. Bacterial growth inhibition was assessed by measuring optical density at 600 nm ($OD_{600}$) over a 48-hour period. For minimal bactericidal concentration (MBC) determination, MRSA was exposed to algae-PNP(Vanc)-robot under identical conditions, and enumeration with the TSB agar (Sigma-Aldrich) plates was performed at 24 h and 48 h post incubation. Control groups included $1 \times$ PBS, free Vanc, PNP(Vanc), and static algae-PNP(Vanc).

### In vivo bactericidal efficiency of algae-PNP(Vanc)-robot

MRSA was initially cultured on a TSB agar plate and incubated at 37 °C for 12 h. A single colony was then inoculated into 10 mL of TSB and grown under agitation at 200 rpm at 37 °C for another 12 h. Bacteria were harvested by centrifugation at $3000 \times g$ for 10 min, washed 3 times with $1 \times$ PBS buffer, and resuspended in $1 \times$ PBS to a final concentration of $2.5 \times 10^8 \, CFU \, mL^{-1}$. Male CD-1 mice were anesthetized with a ketamine-xylazine cocktail and intratracheally inoculated with $1 \times 10^7$ CFU of MRSA. After a 1 h recovery period to ensure normal respiration and movement, the mice were subjected to a 60 min nebulization treatment. Three mice per chamber received either $1 \times$ PBS buffer, algae-PNP(Vanc)-robot, static algae-PNP(Vanc), or PNP(Vanc) at an equivalent Vanc concentration of $10 \, \mu g \, mL^{-1}$. In addition, for comparison with intravenous (IV) therapy, mice received either an equivalent single dose of free Vanc (8.5 μg) or two doses of clinical free Vanc ($30 \, mg \, kg^{-1}$, 12 h per dose) administered IV 2 h post infection. After 24 h, the mice were euthanized, and their lungs were harvested for bacterial load quantification using standard enumeration protocols.

### In vivo survival examination

Male CD-1 mice were put under anesthesia using a ketamine-xylazine cocktail and then intratracheally inoculated with a lethal $5 \times 10^7$ CFU dose of MRSA. Following a 1 h recovery period, the mice underwent a 60 min nebulization treatment. Each chamber contained 3 mice, and the treatments administered included $1 \times$ PBS buffer, algae-PNP(Vanc)-robot, static algae-PNP(Vanc), or PNP(Vanc) at an equivalent Vanc concentration of $10 \, \mu g \, mL^{-1}$. For comparison with IV therapy, one group of mice received a single equivalent dose of free Vanc (8.5 μg), and the other group was given two clinical doses of free Vanc ($30 \, mg \, kg^{-1}$, administered every 12 h). Both IV treatments were initiated 2 h after infection. The mice were monitored daily for survival.

### In vivo biosafety studies

For the in vivo biosafety studies, male CD−1 mice were euthanized on 1, 7, and 14 days following a 60-minute nebulization administration of $1 \times$ PBS buffer or $1 \times 10^8 \, mL^{-1}$ algae-PNP(Vanc)-robot for sample collection. Comprehensive blood chemistry analyses and blood cell counts were performed using serum or whole blood collected in potassium-EDTA tubes (Sarstedt), facilitated by the UCSD Animal Care Program Diagnostic Services Laboratory. For histological examination, major organs were sectioned and stained with haematoxylin and eosin, followed by imaging with a Nanozoomer 2.0-HT slide scanning system (Hamamatsu). All histological assessments were conducted in a blinded manner to prevent observer bias. To examine inflammatory responses, the levels of TNF-α, IL-6, and IL−1β from blood samples after 1, 7, and 14 days post administration were measured with enzyme-linked immunosorbent assay (ELISA) kits (BioLegend).

### Statistical analysis

All experiments were conducted independently and repeated to ensure the reliability of the data presented in the figures. Data are depicted as error bars representing the mean or geometric mean ± standard deviation (s.d.). Statistical significance between two groups was determined using unpaired two-tailed Student's $t$ test, while the significance between more than two groups was analyzed by one-way analysis of variance (ANOVA) followed by Dunnett's test. Significance levels are indicated as follows: NS $P > 0.05$), *$P \le 0.05$, **$P \le 0.01$, ***$P \le 0.001$, and ****$P \le 0.0001$. To mitigate potential bias, no data were excluded, samples were randomly assigned to experimental groups, organisms were cultured under consistent environmental conditions, and randomization was employed in group assignments. Investigators were not blinded during data collection and analysis.

**Reporting summary**

Further information on research design is available in the Nature Portfolio Reporting Summary linked to this article.

## Data availability

All data supporting the findings of this study are available within the article and its supplementary files. Any additional requests for information can be directed to and will be fulfilled by, the corresponding authors. Source data are provided in this paper. All data needed to evaluate the conclusions in the paper are present in the paper or the Supplementary Information. Source data are provided in this paper.

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

## Acknowledgements

This work is supported by the Defense Threat Reduction Agency Joint Science and Technology Office for Chemical and Biological Defense under Grant Numbers HDTRA1-24–1-0019 (J.W.) and HDTRA1-21–1-0010 (L.Z.).

## Author contributions

Z.L., Z.G., F.Z., L.Z., and J.W. conceived the ideas and designed the experiments. Z.L., Z.G., F.Z., and L.S. performed and conducted the experiments. Z.L., Z.G., F.Z., L.S., H.L., Z.F., J.L.D., Y.Z., C.T., A.Z., Y.Y., S.D., D.W., A.C., L.Y., L.M.R., W.G., R.H.F., L.Z., and J.W. discussed and analyzed the data. Z.L., F.Z., R.H.F., L.Z., and J.W. wrote the manuscript. All the authors reviewed, edited, and approved the paper.

## Competing interests

The authors declare that they have no competing interests.
