## [Transparent Peer Review file · Nature Communications]

Inhalable biohybrid microrobots: a non-invasive approach for lung treatment

Corresponding Author: Professor Joseph Wang

Version 0:

Reviewer comments:

Reviewer #1

(Remarks to the Author)

The paper titled "Inhalable Biohybrid Microrobots: A Non-Invasive Approach for Lung Treatment" presents an innovative approach using biohybrid microrobots for pulmonary drug delivery, specifically focusing on a non-invasive method using nebulization. The use of *Micromonas pusilla* algae-based microrobots demonstrates a promising shift from the traditionally invasive intratracheal administration toward more patient-friendly inhalation techniques.

{+} The study does not sufficiently compare the nebulized algae-based biohybrid microrobots to existing lung delivery systems such as liposomes, nanoparticles, or traditional antibiotics via inhalation. A head-to-head comparison with established clinical inhalable therapies would have strengthened the argument for using biohybrid microrobots, particularly with respect to drug distribution, clearance, and therapeutic efficacy.

{+} The nebulization system described is effective for murine models, but the scalability for human use is inadequately addressed. Issues such as aerosol particle size, flow rate, and total inhaled dose are not optimized or tailored for human lungs, which differ significantly in scale and structure. The variability in human lung geometry and lung disease pathophysiology could greatly impact the distribution and retention of the microrobots, which is not accounted for in the study.

{+} The paper acknowledges that algae motility decreases under certain nebulization conditions (e.g., increased loading concentrations or higher flow rates). This instability in motility is a crucial weakness, as active motility is the core advantage of the biohybrid microrobots. However, the authors do not sufficiently explore potential solutions to mitigate this decline or propose alternative methods to ensure consistent motility in various conditions.

{+} The release profile of vancomycin from the algae robots shows that 50% of the drug is released within the first 24 hours, which is relatively rapid. This could lead to a potential mismatch between drug release and therapeutic needs, particularly in chronic conditions where prolonged and controlled release is more beneficial. Furthermore, there is little exploration into how the release rate might be modulated or controlled to match specific clinical needs. This limitation could impact the system's applicability across different lung diseases.

{+} Although some effort was made to test algae motility at body temperature, the study does not investigate how the algae microrobots perform under other physiological or pathological conditions (e.g., different pH levels, mucus production, or inflammatory environments in diseased lungs). The lung is a highly dynamic environment, and the microrobots' ability to maintain motility and stability in varied conditions remains unclear, limiting the certainty of their in vivo performance across different patient populations.

{+} The study does not sufficiently explore the potential for immune response or inflammation induced by algae-based microrobots. While the paper suggests minimal uptake by macrophages and an absence of toxicity in short-term studies, it does not investigate long-term immunogenicity or the possibility of chronic inflammatory responses. Biohybrid systems often introduce the risk of unanticipated immune reactions, which could reduce their long-term safety and efficacy.

{+} The paper focuses on lung retention, but it lacks comprehensive biodistribution studies to assess whether the microrobots might migrate to other organs over time, particularly if they are unintentionally inhaled into the bloodstream or

pass through the alveolar-capillary barrier. The authors should include more detailed pharmacokinetic and biodistribution analysis beyond the lungs to rule out potential off-target effects or systemic toxicity.

Reviewer #2

(Remarks to the Author)

Li et al. have submitted an original research paper titled "Inhalable biohybrid microrobots: a non-invasive approach for lung treatment". The paper is well-written and comprehensive. Yet, several issues need to be addressed before publication. Below are my specific comments in no specific order:

1. The use of the word "innovative" is inappropriate (e.g., see review paper by Zimmermann, Coy J., et al. "Delivery and actuation of aerosolized microrobots." *Nano Select* 3.7 (2022): 1185-1191.). Please revise throughout.
2. The phrases *in vivo*, *in vitro*, etc., should be written in Italics. Please fix it throughout.
3. As a researcher who focuses almost solely on noninvasive drug delivery routes, claiming that (L.53) "Non-invasive administration provides substantial benefits, including effective delivery of therapeutic agents to the respiratory" is simply wrong. That is NOT one of the advantages of noninvasive drug delivery (DD). It is one of its biggest disadvantages compared to injection-based routes. The pulmonary delivery advantage that you can claim could be targeted delivery to the lungs (and thus reduced side effects).
4. L.76, you claim "optimal biodistribution". This means the microrobots resided ONLY in the lungs. I do not see any full-body (and important organs) distribution to prove this claim. Please remove or perform the appropriate experiment to be able to claim so.
5. Regarding the "platelet membrane-coated nanoparticles," it is not clear what the NPs are made of. Please explain.
6. L.60-63, you claim: "However, such static drug carriers commonly encounter various limitations post-inhalation, including inadequate targeting specificity, rapid clearance, and elimination by alveolar macrophages, which compromise their therapeutic efficacy and increase the risk of systemic side effects [ref. 24,25]. The inherent active motility of microrobot drug carriers offers considerable promise for addressing these limitations". And also at L.275 "Unlike static drug carriers, which 275 frequently encounter limitations such as insufficient targeting specificity." - Based on the data you present, the undirected motility only improved clearance issues, but it did not affect targeting. If it did, please explain how and which experiment shows this claim.
7. In addition to comment 6 - the use of platelet membrane coating is probably the reason for increasing resident time (a well-known fact, e.g., see Wei, Xiaoli, et al. "Nanoparticles camouflaged in platelet membrane coating as an antibody decoy for the treatment of immune thrombocytopenia." *Biomaterials* 111 (2016): 116-123.) and NOT the motility. Unless you have an experiment without platelet membrane coating, you cannot claim otherwise.
8. The authors claim that "the algae robot-containing aerosol particles were suspended in an oil phase, and it was observed that they maintained their motility within the droplets (Fig. 1c, Supplementary Fig. 6 and Video 3)." Since you can quantify their motility, please provide the actual numbers of pre and post-nebulization to prove this claim (with proper statistics).
9. Statistical analysis - when performing the student t-test, it was not mentioned whether you performed the normality test priorly. Thus, what did you assume? homoscedastic or heteroscedastic variances?
10. Why only male mice were used (e.g., alveolar macrophage phagocytosis). Nature guidelines require the use of genders equally.
11. When you examine only the lungs, you can not call it simply biodistribution. Biodistribution requires that you test the whole body and important organs such as the liver, kidneys, heart, and brain. Otherwise, change to "lung biodistribution".
12. Units: all units and numbers should be separated by one blank space. Except for Celsius and %. Also, it should be mL and not ml in SI units. Please fix both throughout.
13. The authors write, "it was observed that 73.89 % or 73.77 %...respectively" - first, it should be "and". I assume there were several repetitions, so why not report the average and SD? I find it hard to believe that the accuracy is so high you can even report 100th of a percentage. Please do so to all reported values which do not have an average and SD.
14. Also, when claiming "retained their motility post nebulization", you must at least mention the p-value range obtained to claim this.
15. L.123 "Likewise, algae motility and viability were also negatively impacted." Elaborate and give specific numbers.
16. L.127 "The size distribution of aerosol particles exhibited a pronounced left shift." This is not light interaction detail; just write a decrease in size or something of that sort.
17. "This was reflected in reductions of the average speed, motile ratio, and viability of the algae post nebulization with increasing the system air flow rate." Again, no numbers, and do so everywhere applicable.
18. L.138: "negligible changes in algae robot motion characteristics after nebulization (Fig. 2j)." Define negligible.
19. Please provide an explanation/hypothesis as to why algal bots lose their speed as the temperature increases. Intuitively, this should have been otherwise, no?
20. L.209 "The amount of loaded drug increased proportionally with the algae input" - what is algae input? more algae? algae mass? numbers (the figure just states 10^8 , but of what?)
21. "Subsequently, the minimal inhibitory concentration (MIC) against MRSA was assessed for the algae-PNP(Vanc)-robot alongside different control groups, including free Vanc, PNP(Vanc), and static algae functionalized with PNP(Vanc) (denoted as 'static algae-PNP(Vanc)', Supplementary Fig. 24)." Supp. Figure 24 is an SEM image. Please fix it (it seems like Fig 25). Also, please specify that MIC was determined *in vitro*.
22. "The findings revealed that approximately 8.5 μg could be delivered per lung after 60 min of nebulization with algae-PNP(Vanc)-robot (Fig. 4c). Comparable results were obtained for the static algae-PNP(Vanc) and PNP(Vanc) groups (Supplementary Fig. 30). - why Fig. 30 "free" and was not checked as well?
23. In L.244, free vanc was given IV; why not compare it to free vanc given via nebulization?
24. L.278 "This led to enhanced retention in mouse lungs 278 over an extended period of several days." - Figure 17 in the

Supp file shows that the other groups also remained in the lungs for this period. Please revise and, more importantly, provide a better explanation as to why the algae bots show better efficacy if they deliver the same amount of drug to the lungs as the other groups and reside for about the same duration (based on Fig 17, perhaps since more is visible after 72 h?)

25. Generally, my biggest concern is that the proper control of administering free vanc via nebulization was not done in many places where it was needed (e.g., biodistribution, residence time, and lung uptake). Without these control group experiments, you cannot claim that "Our results confirmed that the inherent long-lasting self-propulsion of the biohybrid microrobots makes them an attractive choice for delivering therapeutic agents to the lungs".

26. L.283 "A thorough evaluation of the biosafety..." For a thorough evaluation, you will have to do full biodistribution, especially since you have nanoparticles. Remove the word if you do not plan to do so.

27. Why were biodistribution studies not performed with PNP (Fig 3)? This is the actual drug carrier...

27. Regarding figures, please try to increase font size where possible.

Reviewer #3

(Remarks to the Author)

This study investigates the use of inhalable bio-hybrid microrobots in pneumonia research. In contrast to earlier studies (Nat. Mater. 21, 1324–1332 (2022) and Sci. Adv. 10, eadn6157 (2024)), this research introduces a novel approach for the non-invasive treatment of lung diseases, demonstrating both innovative and practical significance in terms of technology and methodology. The paper is well-structured, and the experimental data presented is extensive; however, there is a need for a more thorough analysis, as well as a deeper discussion of the experimental results and data.

1. Vancomycin can be directly administered via inhalation, and the author should add a corresponding group.

2. The author claims in the article that "Long term survival rates for mice treated with PNP (Vanc) and clinical IV doses of free Vanc were 16.67% and 66.67%, respectively." Vancomycin, recognized as a targeted treatment for drug-resistant MRSA, exhibits a cure rate of merely 66.67%, which is quite astonishing. It is imperative for the author to present official or high-quality clinical research studies that encompass a substantial amount of data.

3. In the treatment of MRSA, the dosage of medication significantly influences the therapeutic results. The author has not standardized the dosages for intravenous administration and robot-assisted delivery, raising questions about how can conclude that robot-assisted delivery yields superior treatment effects at equivalent dosages.

4. The author claimed the drug delivery mechanism of this algae as a robotic system, intending to illustrate that the algae's mobility allows it to avoid phagocytosis by macrophages, thus improving the drug's retention in the lungs. To substantiate this perspective, the author is required to conduct in vitro experiments, which are essential for validating the efficacy of the robotic system.

5. Some images are reused in the main text and supplementary materials

6. The author should provide analysis of the drugs release process by robots to determine the correlation between the robot's residency effect and therapeutic efficacy.

Version 1:

Reviewer comments:

Reviewer #1

(Remarks to the Author)

The authors have addressed all my comments in their revised manuscript. The manuscript has been improved and therefore I would like to recommend it for publication.

Reviewer #2

(Remarks to the Author)

Dear Authors, thank you for the revisions made. Please see additional comments to your specific replies.

1. Regarding comment#8 - OK, I can accept that visualizing would not be indicative of the true diffusion rate. However, my point was that you claimed that "the algae robot-containing aerosol particles were suspended in an oil phase, and it was observed that they maintained their motility within the droplets". If you say the motility was "maintained" then you had to perform some kind of quantitative analysis to be able to claim that. If you did not and it was only observed visually without quantification then you should rephrase to "seem to maintain".

2. Comment #10 - Well, at the very least please add this explanation and hypothesis to the paper itself. As for accepting

whether it is ok to use only male rats, I leave to the editor to decide as this is part of Nature's own guidelines.

3. Comment #15: please remember significant digits. So writing "from $58.46 \pm 6.42 \mu\text{m s}^{-1}$ to $45.46 \pm 9.87 \mu\text{m s}^{-1}$ " if the SD is already 6.42 you cannot be accurate up to 58.46. Better write is as 58 ± 6.4 . Do the same for the other pair of AVG and SD and throughout if needed.

4. Comment #23&25: OK, I can accept this reasoning. In this case, please add what you hypothesize will happen to free vanc when nebulized in order to emphasize why you need you delivery system (and since you chose not to compare to it).

As these further revisions are minor, I see no reason to re-review them and incur further delays to the authors.

Reviewer #3

(Remarks to the Author)

The authors have made modifications according to the review comments.

Detailed List of CHANGES and Responses to Reviewers' Comments:

Reviewer 1

General Comment: *The paper titled "Inhalable Biohybrid Microrobots: A Non-Invasive Approach for Lung Treatment" presents an innovative approach using biohybrid microrobots for pulmonary drug delivery, specifically focusing on a non-invasive method using nebulization. The use of *Micromonas pusilla* algae-based microrobots demonstrates a promising shift from the traditionally invasive intratracheal administration toward more patient-friendly inhalation techniques.*

Response: We appreciate the Reviewer for acknowledging the novelty and impact of our approach. We have addressed the Reviewer's comments below point by point.

Comment #1: *The study does not sufficiently compare the nebulized algae-based biohybrid microrobots to existing lung delivery systems such as liposomes, nanoparticles, or traditional antibiotics via inhalation. A head-to-head comparison with established clinical inhalable therapies would have strengthened the argument for using biohybrid microrobots, particularly with respect to drug distribution, clearance, and therapeutic efficacy.*

Response #1: We appreciate the Reviewer's feedback. We agree on the value of a comprehensive comparison with established clinical therapies. However, since intravenous (IV) vancomycin delivery remains the standard in clinical settings, it was included as part of our study to demonstrate the clinical relevance of our formulation (**Fig. 4d,e**). Inhalable free drug vancomycin is not used clinically, so we chose not to include it as a control. Instead, we used nebulized vancomycin-loaded PNP nanoparticles (PNP(Vanc)) as a control group (**Fig.4d,e**), providing a more relevant comparison to our nebulized algae-PNP(Vanc)-robot formulation.

Comment #2: *The nebulization system described is effective for murine models, but the scalability for human use is inadequately addressed. Issues such as aerosol particle size, flow rate, and total inhaled dose are not optimised or tailored for human lungs, which differ significantly in scale and structure. The variability in human lung geometry and lung disease pathophysiology could greatly impact the distribution and retention of the microrobots, which is not accounted for in the study.*

Response #2: We thank the Reviewer for the valuable comment. Our proof-of-concept study highlights the advantages of the active algae-based drug delivery in small animal models. Translating this innovative nebulization approach to human applications will require scalable adaptations that account for the unique structural and dimensional characteristics of human lungs, ensuring the efficient distribution and retention achieved in preclinical models. Although this study primarily establishes feasibility in murine models, we acknowledge the critical translational challenges ahead. To address these, we have expanded the discussion in the revised manuscript

(pages 15-16), outlining considerations for scalability and potential parameter adjustments, such as total dosage and aerosol particle size, to optimize delivery for human lung physiology.

Comment #3: The paper acknowledges that algae motility decreases under certain nebulization conditions (e.g., increased loading concentrations or higher flow rates). This instability in motility is a crucial weakness, as active motility is the core advantage of the biohybrid microrobots. However, the authors do not sufficiently explore potential solutions to mitigate this decline or propose alternative methods to ensure consistent motility in various conditions.

Response #3: We thank the Reviewer for the valuable suggestion. While the flow velocity from the nebulization system impacts the algae microrobot velocity to a certain degree (~20%), the microrobots still maintain sufficient propulsion and speed over extended periods to escape the macrophage cells and achieve long-term retention. To further evaluate the motility strength of post-nebulized algae-based microrobots, we conducted new *in vitro* experiments to compare their ability to escape from the immune cell uptake. It was shown that the post-nebulized algae robot behaved similarly as the algae robot prior to nebulization. The description of post-nebulization velocities was included on **page 9** and the new experimental data were added in the new **Supplementary Fig. 19**.

Comment #4: The release profile of vancomycin from the algae robots shows that 50% of the drug is released within the first 24 hours, which is relatively rapid. This could lead to a potential mismatch between drug release and therapeutic needs, particularly in chronic conditions where prolonged and controlled release is more beneficial. Furthermore, there is little exploration into how the release rate might be modulated or controlled to match specific clinical needs. This limitation could impact the system's applicability across different lung diseases.

Response #4: We appreciate the Reviewer for bringing this important point to our attention. It should be emphasized that the main goal of our study was to demonstrate the advantages of our newly developed method for active biohybrid delivery to the lungs via nebulization. While the platelet membrane-coated PLGA nanoparticles provide rapid release which satisfies the therapeutic needs for our studies, it holds the potential for alternative cargos that are suitable for slower drug release. We have now added additional discussion on this topic to the revised manuscript (**page 16**).

Comment #5: Although some effort was made to test algae motility at body temperature, the study does not investigate how the algae microrobots perform under other physiological or pathological conditions (e.g., different pH levels, mucus production, or inflammatory environments in diseased lungs). The lung is a highly dynamic environment, and the microrobots' ability to maintain motility and stability in varied conditions remains unclear, limiting the certainty of their in vivo performance across different patient populations.

Response #5: We thank the Reviewer for raising this important comment. To provide more insight into algae motility under various realistic lung environments, we performed several new *in vitro* experiments to test the performance of the algae robots in the presence of mucus and cytokines at pH=6 at 37 °C. The new results were included in the new **Supplementary Fig. 14**, and the related discussion was added on **page 8**.

Comment #6: The study does not sufficiently explore the potential for immune response or inflammation induced by algae-based microrobots. While the paper suggests minimal uptake by macrophages and an absence of toxicity in short-term studies, it does not investigate long-term immunogenicity or the possibility of chronic inflammatory responses. Biohybrid systems often introduce the risk of unanticipated immune reactions, which could reduce their long-term safety and efficacy.

Response #6: We thank the Reviewer for raising this question regarding long-term immunogenicity. To address this concern, we conducted additional experiments to assess potential inflammatory responses by measuring pro-inflammatory cytokine levels in mice following nebulization of algae-based microrobots. The serum cytokine levels were measured on days 1, 7 and 14 post-administration and the results showed that no discernable systemic inflammation was induced by the microrobots. The new data were included in the new **Supplementary Fig. 33** and the corresponding description was added to **page 14**.

Comment #7: The paper focuses on lung retention, but it lacks comprehensive biodistribution studies to assess whether the microrobots might migrate to other organs over time, particularly if they are unintentionally inhaled into the bloodstream or pass through the alveolar-capillary barrier. The authors should include more detailed pharmacokinetic and biodistribution analysis beyond the lungs to rule out potential off-target effects or systemic toxicity.

Response #7: We thank the Reviewer for highlighting the importance of assessing potential off-target migration and systemic exposure. Unlike smaller nanoparticles, our biohybrid microrobots are engineered at a size greater than 1 μm , which inherently restricts their ability to cross the alveolar-capillary barrier and thus limits their distribution to lung tissue. This size characteristic not only enhances their localized retention within the lungs but also minimizes the likelihood of migration to other organs, reducing the risk of systemic exposure. Our study's focus on lung retention reflects this localized therapeutic strategy. We recognize, however, the value of future validation studies to further characterize the biodistribution and confirm this safety profile.

Reviewer 2

General Comment: *Li et al. have submitted an original research paper titled "Inhalable biohybrid microrobots: a non-invasive approach for lung treatment". The paper is well-written*

and comprehensive. Yet, several issues need to be addressed before publication. Below are my specific comments in no specific order:

Response: We thank the Reviewer for recognizing the originality and comprehensiveness of our study, and for the many useful comments and suggestions towards improving our manuscript. Below we have responded to all the comments point by point.

Comment #1: *The use of the word "innovative" is inappropriate (e.g., see review paper by Zimmermann, Coy J., et al. "Delivery and actuation of aerosolized microbots." Nano Select 3.7 (2022): 1185-1191.). Please revise throughout.*

Response #1: We thank the Reviewer for the useful suggestion. Accordingly, we deleted the word 'innovative' throughout the text. Note that Zimmerman's aerosol work did not include information on biohybrid algae-based microrobot with active motility.

Comment #2: *The phrases *in vivo*, *in vitro*, etc., should be written in Italics. Please fix it throughout.*

Response #2: Following the Reviewer's suggestion, we rewrote all these phrases in *italics* throughout the manuscript.

Comment #3: *As a researcher who focuses almost solely on noninvasive drug delivery routes, claiming that (L.53) "Non-invasive administration provides substantial benefits, including effective delivery of therapeutic agents to the respiratory" is simply wrong. That is NOT one of the advantages of noninvasive drug delivery (DD). It is one of its biggest disadvantages compared to injection-based routes. The pulmonary delivery advantage that you can claim could be targeted delivery to the lungs (and thus reduced side effects).*

Response #3: We thank the Reviewer for pointing out this oversight. As correctly stated, a major advantage of non-invasive drug delivery is its ability to reduce side effects, rather than effective delivery to the respiratory system. We have revised the text accordingly on **page 3** to reflect this point, emphasizing that the key benefit of the non-invasive administration would be targeted delivery to the lungs.

Comment #4: *L.76, you claim "optimal **biodistribution**". This means the microbots resided **ONLY in the lungs**. I do not see any full-body (and important organs) distribution to prove this claim. Please remove or perform the appropriate experiment to be able to claim so.*

Response #4: We thank the Reviewer for the comment. We have removed the term in question and revised the language on **page 4** to more accurately reflect the findings.

Comment #5: Regarding the "platelet membrane-coated nanoparticles," it is not clear what the NPs are made of. Please explain.

Response #5: We apologize for the confusion. The nanoparticle cores are made of poly (lactic-co-glycolic acid) (PLGA) and are coated with platelet membranes. We have clarified this in the revised manuscript and added the precise term "PLGA" to the relevant section on **page 4**.

Comment #6: L.60-63, you claim: "However, such static drug carriers commonly encounter various limitations post-inhalation, including inadequate targeting specificity, rapid clearance, and elimination by alveolar macrophages, which compromise their therapeutic efficacy and increase the risk of systemic side effects [ref. 24,25]. The inherent active motility of microrobot drug carriers offers considerable promise for addressing these limitations". And also at L.275 "Unlike static drug carriers, which 275 frequently encounter limitations such as insufficient targeting specificity."- Based on the data you present, the undirected motility only improved clearance issues, but it did not affect targeting. If it did, please explain how and which experiment shows this claim.

Response #6: We thank the Reviewer for the valuable comment. The phrases in question on **pages 4 and 14** have been removed to avoid potential misleading information. As such, the text has been revised to focus on the microrobots' advantage of delayed clearance and macrophage escape, which aligns with the experimental data provided. We trust that this change provides greater clarity and more accurately describes the therapeutic benefits of microrobot drug carriers.

Comment #7: In addition to comment 6 - the use of platelet membrane coating is probably the reason for increasing resident time (a well-known fact, e.g., see Wei, Xiaoli, et al. "Nanoparticles camouflaged in platelet membrane coating as an antibody decoy for the treatment of immune thrombocytopenia." *Biomaterials* 111 (2016): 116-123.) and NOT the motility. Unless you have an experiment without platelet membrane coating, you cannot claim otherwise.

Response #7: We thank the Reviewer for raising this important question. The platelet membrane-coated nanoparticles were identically applied in both the active algae group and the static algae group. A comparison between the two groups shows a clear advantage in retention time under the influence of enhanced motility.

Comment #8: The authors claim that "the algae robot-containing aerosol particles were suspended in an oil phase, and it was observed that they maintained their motility within the droplets (Fig. 1c, Supplementary Fig. 6 and Video 3)." Since you can quantify their motility, please provide the actual numbers of pre and post-nebulization to prove this claim (with proper statistics).

Response #8: We thank the Reviewer for the useful comment. Oil droplets were employed to enhance the visualization of algae motion and to show the successful encapsulation of the microrobots into aerosol produced by the nebulizer system. However, tracking the algae within

such a confined space would be inaccurate with our tracking method. The microrobots' motility is measured post-nebulization after they reach a stable solution, as discussed on page 6, and a corresponding description indicating the purpose of oil phase visualization was added to **page 5**.

***Comment #9:** Statistical analysis - when performing the **student t-test**, it was not mentioned whether you performed the normality test priorly. Thus, what did you assume? homoscedastic or heteroscedastic variances?*

Response #9: We thank the Reviewer for the useful suggestion. We revised the statistical analysis description in the **Methods** part on **Page 33** and the corresponding figure's descriptions. We performed the normality test before we used unpaired two-tailed student's *t*-test via GraphPad Prism 10. We assumed that all data sets following Gaussian distribution and homoscedastic variances.

***Comment #10:** Why only male mice were used (e.g., alveolar macrophage phagocytosis). Nature guidelines require the use of genders equally.*

Response #10: We thank the Reviewer for the comment regarding the inclusion of both genders in preclinical studies. For this proof-of-concept study, male mice were selected to maintain experimental consistency and reduce potential variables, such as hormonal influences on alveolar macrophage activity, which could introduce variability unrelated to our primary objectives. Given that the primary aim of our study was to establish the feasibility and retention of biohybrid microrobots in lung tissue, we do not anticipate that gender differences would significantly impact these foundational results.

***Comment #11:** When you examine **only the lungs**, you cannot call it simply biodistribution. Biodistribution requires that you test the whole body and important organs such as the liver, kidneys, heart, and brain. Otherwise, change to "lung biodistribution".*

Response #11: We thank the Reviewer for raising this valuable suggestion. Accordingly, the text was revised to "lung biodistribution" on **pages 2 and 4** to ensure the accuracy of our description.

***Comment #12:** Units: all units and numbers should be separated by one blank space. Except for Celcius and %. Also, it should be mL and not ml in SI units. Please fix both throughout.*

Response #12: We thank the Reviewer for pointing out this format standardization. The text was revised to separate all units and numbers by one blank space. The SI units of milliliter were revised to be mL, while the microliter was revised to μL . We made these changes throughout the manuscript and SI.

Comment #13: *The authors write, "it was observed that 73.89 % or 73.77 %...respectively" - first, it should be "and". I assume there were several repetitions, so why not report the average and SD? I find it hard to believe that the accuracy is so high you can even report 100th of a percentage. Please do so to all reported values which do not have an average and SD. (L108-113)*

Response #13: We thank the Reviewer for the valuable comment. The text on **page 6** was revised to $73.89 \pm 3.18 \%$ and $73.77 \pm 2.94 \%$ on **page 6**, indicating the averaged values and SD. For assessing the motility and viability of biohybrid microrobots, we typically calculate these metrics from motion videos and microscopy images. For instance, in one sample, we observed 23 static algae among 423 active algae-based biohybrid microrobots. For the percentage calculation, we kept two decimal places. We have now included the standard deviation (SD) to enhance the scientific rigor and accuracy of our results.

Comment #14: *Also, when claiming "retained their motility post nebulization", you must at least mention the p-value range obtained to claim this.*

Response #14: We thank the Reviewer for the valuable comment. Accordingly, we added the p-value on **page 6** for this description on post-nebulization motility.

Comment #15: *L.123 "Likewise, algae motility and viability were also negatively impacted." Elaborate and give specific numbers.*

Response #15: We thank the Reviewer for the valuable comment. The text was revised to mention the numerical speed data (from $58.46 \pm 6.42 \mu\text{m s}^{-1}$ to $45.46 \pm 9.87 \mu\text{m s}^{-1}$) on **page 6**.

Comment #16: *L.127 "The size distribution of aerosol particles exhibited a pronounced left shift." This is not light interaction detail; just write a decrease in size or something of that sort.*

Response #16: We apologize for the confusion. To most accurately describe the change in aerosol sizes, the precise numerical values were added to the specific sentence on **page 7**.

Comment #17: *"This was reflected in reductions of the average speed, motile ratio, and viability of the algae post nebulization with increasing the system air flow rate." Again, no numbers, and do so everywhere applicable.*

Response #17: We thank the Reviewer for this useful comment. Numerical values for the speed range, motility ratio and viability have been added to the revised **page 7**.

Comment #18: *L.138: "negligible changes in algae robot motion characteristics after nebulization (Fig. 2j)." Define negligible.*

Response #18: We thank the Reviewer for this useful comment. To avoid confusion, we have updated the language to indicate that similar distances and patterns of travel were observed when comparing pre-nebulization and post-nebulization trajectories shown in Figure 2j. The text was revised on **page 7** to address this comment.

Comment #19: Please provide an explanation/hypothesis as to why algal bots lose their speed as the temperature increases. Intuitively, this should have been otherwise, no?

Response #19: We thank the Reviewer for the valuable comment. The optimal culture condition for the algae species *Micromonas pusilla* is at room temperature. Increasing the temperature to body temperature leads to a mild suppression in metabolism. However, despite the reduction, their motility remains effective even under body temperature, as demonstrated in our study. An explanation has been added to **page 8**.

Comment #20: L.209 "The amount of loaded drug increased proportionally with the algae input" - what is algae input? more algae? algae mass? numbers (the figure just states 10⁸, but of what?

Response #20: We apologize for the confusion. It's with the increase of algae number. This has now been updated in the revised manuscript on **page 11**.

Comment #21: "Subsequently, the minimal inhibitory concentration (MIC) against MRSA was assessed for the algae-PNP(Vanc)-robot alongside different control groups, including free Vanc, PNP(Vanc), and static algae functionalized with PNP(Vanc) (denoted as 'static algae-PNP(Vanc)', Supplementary Fig. 24)." Supp. Figure 24 is an SEM image. Please fix it (it seems like Fig 25). Also, please specify that MIC was determined in vitro.

Response #21: We thank the Reviewer for pointing out this potential confusion. The former Supplementary Fig. 24, now Supplementary Fig. 26, displays an SEM image of a static algae-PNP(Vanc)-robot, where the missing flagella indicates a loss of motility. Following this comment, we added to **page 11** the description “*The static algae-PNP(Vanc) was characterized by SEM imaging, which revealed the absence of flagella that are required for motility (Supplementary Fig. 26)*” to address this potential confusion. In addition, the description was also added on **page 11** to specify that the MIC study was conducted *in vitro*.

Comment #22: "The findings revealed that approximately 8.5 µg could be delivered per lung after 60 min of nebulization with algae-PNP(Vanc)-robot (Fig. 4c). Comparable results were obtained for the static algae-PNP(Vanc) and PNP(Vanc) groups (Supplementary Fig. 30). - why Fig. 30 "free" and was not checked as well?

Response #22: We thank the Reviewer for this valuable comment. In our study, free vancomycin was selected as a control group and administered through intravenous injection, which we

considered the most appropriate comparison due to its use in the clinic. Therefore, Supplementary Fig. 32 (formerly Supplementary Fig. 30) includes the two key groups: static algae-PNP(Vanc) and PNP(Vanc). By demonstrating that the amount of payload delivered was comparable between all groups, this enabled us to validate the importance of using motile algae for our platform.

Comment #23: *In L.244, free vanc was given IV; why not compare it to free vanc given via nebulization?*

Response #23: We greatly appreciate the Reviewer's feedback on nebulization of free vancomycin. Since intravenous (IV) vancomycin delivery remains the standard in clinical settings, it was included as part of our study to demonstrate the clinical relevance of our formulation (**Fig. 4d,e**). Inhalable free drug vancomycin is not used clinically, so we chose not to include it as a control. This has now been clarified in the revised manuscript on **page 13**.

Comment #24: *L.278 "This led to enhanced retention in mouse lungs over an extended period of several days." - Figure 17 in the Supp file shows that the other groups also remained in the lungs for this period. Please revise and, more importantly, provide a better explanation as to why the algae bots show better efficacy if they deliver the same amount of drug to the lungs as the other groups and reside for about the same duration (based on Fig 17, perhaps since more is visible after 72 h?)*

Response #24: We thank the Reviewer for the valuable suggestion. In the first 24 hours, the algae robots demonstrated superior retention compared to the static robot control group, as shown in Supplementary Fig. 18 (formerly Supplementary Fig. 17). We would like to clarify that the intratracheal method served as a control to confirm that our nebulizer delivery method did not affect the motility of the algae robots. We believe the longer retention of the algae robots in the lungs will deliver more drugs than the static algae, which were taken up and cleared by the immune cells more quickly. In fact, the therapeutic efficacy in Fig. 4d,e validated this hypothesis. A corresponding discussion has been added on **page 10** for further clarification.

Comment #25: *Generally, my biggest concern is that the proper control of administering free vanc via nebulization was not done in many places where it was needed (e.g., biodistribution, residence time, and lung uptake). Without these control group experiments, you cannot claim that "Our results confirmed that the inherent long-lasting self-propulsion of the biohybrid microrobots makes them an attractive choice for delivering therapeutic agents to the lungs".*

Response #25: We appreciate the Reviewer's feedback. In terms of the selection of proper control groups, since intravenous (IV) vancomycin delivery remains the standard in clinical settings, it was included as part of our study to demonstrate the clinical relevance of our formulation (**Fig. 4d,e**). Inhalable free drug vancomycin is not used clinically, so we chose not to include it as a control. Instead, we used nebulized vancomycin-loaded PNP nanoparticles (PNP(Vanc)) as a

control group (**Fig.4d,e**), providing a more relevant comparison to our nebulized algae-PNP(Vanc)-robot formulation. This has been added to the revised manuscript on **Page 13**.

Comment #26: L.283 "A thorough evaluation of the biosafety..." For a thorough evaluation, you will have to do full biodistribution, especially since you have nanoparticles. Remove the word if you do not plan to do so.

Response #26: We thank the Reviewer for the valuable comment. We have revised the statement on **page 14** and removed the word for a more accurate reflection of the findings.

Comment #27: Why were biodistribution studies not performed with PNP (Fig 3)? This is the actual drug carrier...

Response #27: We appreciate the Reviewer's comment regarding the inclusion of biodistribution studies for the PNPs. In this proof-of-concept study, our main objective was to demonstrate the feasibility of algae-based biohybrid microrobots for lung-targeted delivery, focusing on their motility and retention to enhance drug efficacy. Given this focus, we employed non-motile (static) algae as the main control to assess the impact of active motility on retention. The PNPs, in contrast, differ significantly from the biohybrid microrobots in size, composition, and their potential to cross biological barriers, making them less suitable as a control in our lung biodistribution assessment. A corresponding explanation to address this concern has been included on **page 9**.

Comment #28: Regarding figures, please try to increase font size where possible.

Response #28: We have increased the font size of Fig. 3 and Fig. 4 to provide a better view for readers. The changes can be seen on **pages 39 and 41**.

Reviewer 3

General Comment: *This study investigates the use of inhalable bio-hybrid microrobots in pneumonia research. In contrast to earlier studies (Nat. Mater. 21, 1324–1332 (2022) and Sci. Adv. 10, eadn6157 (2024)), this research introduces a novel approach for the non-invasive treatment of lung diseases, demonstrating both innovative and practical significance in terms of technology and methodology. The paper is well-structured, and the experimental data presented is extensive; however, there is a need for a more thorough analysis, as well as a deeper discussion of the experimental results and data.*

Response: We thank the Reviewer for these encouraging and constructive comments and for recognizing our work as well-structured and extensive.

Comment #1: *Vancomycin can be directly administered via inhalation, and the author should add a corresponding group.*

Response #1: We thank the Reviewer for this useful comment. In terms of the selection of proper control groups, since intravenous (IV) vancomycin delivery remains the standard in clinical settings, it was included as part of our study to demonstrate the clinical relevance of our formulation (**Fig. 4d,e**). Inhalable free drug vancomycin is not used clinically, so we chose not to include it as a control. A corresponding clarification about this was added to **page 13**.

Comment #2: *The author claims in the article that "Long term survival rates for mice treated with PNP (Vanc) and clinical IV doses of free Vanc were 16.67% and 66.67%, respectively." Vancomycin, recognized as a targeted treatment for drug-resistant MRSA, exhibits a cure rate of merely 66.67%, which is quite astonishing. It is imperative for the author to present official or high-quality clinical research studies that encompass a substantial amount of data.*

Response #2: We thank the Reviewer for raising this question. Firstly, although vancomycin is a well-established treatment for drug-resistant MRSA in humans, our study employs an acute MRSA-induced pneumonia model in mice. Consequently, the therapeutic outcomes observed in this preclinical mouse model were not directly translatable to human clinical results. Furthermore, according to previous works [*Pediatric research* 65, 420-424 (2009)] on clinical IV doses of free vancomycin in murine model treatment, a 15 mg kg⁻¹ of IV dosed vancomycin achieved near 75% survival rate on 48 h for neonatal mice. Another study [*Antimicrobial agents and chemotherapy* 46, 3288-3291 (2002)] indicated that at a dosage of 100 mg kg⁻¹ IV dosed vancomycin, mice survival rate remained stable up to 72 h, and stated a 45% survival rate on Day 10. Hence, the 30 mg kg⁻¹ dosage amount we applied in our study was receiving a reasonable outcome with a survival rate of 66.67% at Day 60 (n=6), and served well with the purpose of illustrating the reduced requirement on dosage amount for algae-based microrobot treatment control group.

Comment #3: *In the treatment of MRSA, the dosage of medication significantly influences the therapeutic results. The author has not standardised the dosages for intravenous administration and robot-assisted delivery, raising questions about how to conclude that robot-assisted delivery yields superior treatment effects at equivalent doses.*

Response #3: We thank the Reviewer for the valuable suggestion. In our survival study, shown in **Fig. 4e**, the algae robots administered via nebulization delivered a total vancomycin dose of 8.5 µg per mouse. For comparison, a standardized dose of vancomycin via intravenous (IV) injection is also presented in the same figure panel. It is important to highlight that the clinical IV treatment dose is 30 mg kg⁻¹, equivalent to 600 µg per mouse. These results demonstrate that the algae robots achieved higher treatment efficacy with a lower dose compared to conventional IV treatment. To clarify the dosage used for IV injection, we have added this discussion on the revised **page 13**.

***Comment #4:** The author claimed the drug delivery mechanism of this algae as a robotic system, intending to illustrate that the algae's mobility allows it to avoid phagocytosis by macrophages, thus improving the drug's retention in the lungs. To substantiate this perspective, the author is required to conduct in vitro experiments, which are essential for validating the efficacy of the robotic system.*

Response #4: Following the Reviewer's suggestion, we conducted new in vitro experiments to analyze the immune uptake of pre-nebulized and post-nebulized algae robots and a corresponding static algae formulation. The results clearly showed that the signal of static algae post-nebulization almost disappeared within 24 h, indicating rapid uptake by macrophages in a short time in vitro. In contrast, the signals of algae robot pre-nebulization and post-nebulization disappeared at 72 h, indicating that the motility of algae robots enables them to avoid phagocytosis by macrophages in vitro. The data were added to **Supplementary Fig. 19** and discussed on **page 9**.

***Comment #5:** Some images are reused in the main text and supplementary materials.*

Response #5: We thank the Reviewer for the comment. We have carefully checked all the supplementary figures and the main figures and found no figures being reused. If the Reviewer could help to point out specific figures/panels that are questionable, we would be more than happy to double check again.

***Comment #6:** The author should provide analysis of the drug release process by robots to determine the correlation between the robot's residency effect and therapeutic efficacy.*

Response #6: We thank the Reviewer for the valuable comment. In Supplementary Fig. 25 (formerly Supplementary Fig. 23), the cumulative drug release profile of PNP(Vanc) and algae-PNP(Vanc) was quantified from 0 h to 96 h. At 72 h, the vancomycin was shown to be released 80%, achieving a corresponding therapeutic effect. The static algae microrobot group with no residency effect was shown to be removed by the lung clearance mechanism, as the signal was significantly lower than active algae groups at 24 h. Hence, their therapeutic efficacy was largely hampered due to insufficient drug release prior to clearance.